# MONOTONICITY AS A REQUIREMENT AND AS A REGULARIZER: EFFICIENT METHODS AND APPLICATIONS

## ABSTRACT

We study the setting where risk minimization is performed over general classes of models and consider two cases where monotonicity is treated as either a requirement to be satisfied everywhere or a useful property. We specifically consider cases where point-wise gradient penalties are used alongside the empirical risk during training. In our first contribution, we show that different choices of penalties define the regions of the input space where the property is observed. As such, previous methods result in models that are monotonic only in a small volume of the input space. We thus propose an approach that uses mixtures of training instances and random points to populate the space and enforce the penalty in a much larger region. As a second contribution, we introduce the notion of monotonicity as a regularization bias for convolutional models. In this case, we consider applications, such as image classification and generative modeling, where monotonicity is not a hard constraint but can help improve some aspects of the model. Namely, we show that using group monotonicity can be beneficial in several applications such as: (1) defining strategies to detect anomalous data, (2) allowing for controllable data generation, and (3) generating explanations for predictions. Our proposed approaches do not introduce relevant computational overhead while leading to efficient procedures that provide extra benefits over baseline models.

## 1 INTRODUCTION

Highly expressive model classes such as artificial neural networks have achieved impressive prediction performance across a broad range of supervised learning tasks and domains (Krizhevsky et al., 2012; Graves & Jaitly, 2014; Bahdanau et al., 2014). However, finding predictors attaining low risk on unseen data is often not enough to enable the use of such models in practice. In fact, practical applications usually have more requirements other than prediction accuracy. Hence, devising approaches that search risk minimizers satisfying practical needs led to several research threads seeking to enable the use of neural networks in *real-life* scenarios. Examples of such requirements include: (1) *Robustness*, where low risk is expected even if the model is evaluated under distribution shifts, (2) *Fairness*, where the performance of the model is expected to not significantly change across different sub-populations of the data, and (3) *Explainability/Interpretability*, where models are expected to indicate how the features of the data affect their predictions.

In addition to the requirements mentioned above, a property commonly expected in trained models in certain applications is *monotonicity* with respect to some subset of the input dimensions, i.e., an increase (or decrease) along some particular dimensions strictly imply the function value will not decrease (or will not increase), provided that all other dimensions are kept fixed. As a result, the behavior of monotonic models will be more aligned with the properties that the data under consideration is believed to satisfy. For example, in the models used to accept/reject job applications, we expect acceptance scores to be monotonically non-decreasing with respect to features such as past years of experience of a candidate. Thus, given two applicants with exactly the same features except their years of experience, the more experienced candidate should be assigned an equal or higher chance of getting accepted. For applications where monotonicity is expected, having a predictor failing to satisfy this requirement would damage the user's confidence. As such, different strategies have been devised in order to enable training monotonic predictors. These approaches can be divided into two main categories. The first one is *monotonicity by construction*, where the focus lies on defining a model class that guarantees monotonicity in all of its elements Bakst et al. (2021);

Wehenkel & Louppe (2019); Nguyen & Martínez (2019); You et al. (2017); Garcia & Gupta (2009); Archer & Wang (1993). However, this approach can not be used with general architectures in the case of neural networks. Additionally, the model class might be so constrained that it might affect the prediction performance. Alternatively, a second approach is based on *encouraging monotonicity during training*, i.e., searching for monotonic candidates within a general class of models (Liu et al., 2020; Sivaraman et al., 2020; Gupta et al., 2019). Such a group of methods is more generally applicable and can be used, for instance, with any neural network architecture. However, they are not guaranteed to yield monotonic predictors unless extra verification/certification steps are performed, which can be computationally very expensive. In addition to being a *requirement* as in the examples discussed above, monotonicity has also been observed to be a useful feature in certain cases. For example, it can define an effective inductive bias and improve generalization in cases where prior knowledge indicates the data generation process satisfies such a property (Dugas et al., 2001). In such cases, however, it is not necessary to satisfy the property everywhere, since it is enforced simply as a desirable *feature* of trained models rather than a design specification.

In this work, we tackle the problem of performing empirical risk minimization over rich classes of models such as neural networks, while simultaneously searching for monotonic predictors within the set of risk minimizing solutions. In summary, our contributions are two-fold:

1. *monotonicity as a requirement*: We show that previous methods can only satisfy monotonicity either near the training data or near the boundaries of the input domain. Then, we propose an efficient algorithm that tackles this problem. In short, we apply Mixup (Zhang et al., 2018) between the data and random points to populate the input space, which is shown to enforce monotonicity in a larger volume relative to previous methods in literature. To the best of our knowledge, this is the first work that studies the effect of the sample points used in calculating the monotonicity constraints.

2. *monotonicity as a regularizer*: We define a new notion of monotonicity which is shown to be useful when enforced for applications such as object recognition or generative modeling of images. In these cases, it is not necessarily required to enforce the property everywhere and, as such, the constraints that focus only on the actual data points are discussed (i.e., mixup is not needed). Models satisfying the property are compared with standard predictors across three applications, where it is shown the property is beneficial without compromising the original performance.

## 2 BACKGROUND AND RELATED WORK

We start by defining the notion of *partial monotonicity* used throughout the paper. Consider the standard supervised learning setting where data instances are observed in pairs $x, y \sim \mathcal{X} \times \mathcal{Y}$, where $\mathcal{X} \subset \mathbb{R}^d$ and $\mathcal{Y} \subset \mathbb{R}$ correspond to the input and output spaces, respectively. Further, consider the function $f : \mathcal{X} \mapsto \mathcal{Y}$, and let $M$ indicate some subset of the input dimensions, i.e., $M \subset \{1, ...d\}$, such that $x = \text{concat}(x_M, x_{\bar{M}})$, where $\bar{M} = \{1, ..., d\} \setminus M$. We further overload the notation of function calls to $f$ such that $f(x) = f(x_M, x_{\bar{M}})$.

**Definition 1** *Partially monotonic functions relative to $M$:* We say $f$ is monotonically non-decreasing relative to $M$[1], denoted $f_M$, if $f(x_M, x_{\bar{M}}) \leq f(x'_M, x_{\bar{M}})$, $\forall\, x_M \leq x'_M$, $\forall\, x_{\bar{M}}$, where the comparison $x_M \leq x'_M$ is performed for every dimension.

This definition covers functions that do not decrease in value given increasing changes along a subset of the input dimensions, provided that all other dimensions are kept unchanged. Several approaches were introduced for defining model classes that have such a property. The simplest approach restricts the weights of the network to be non-negative (Archer & Wang, 1993). However, doing so affects the prediction performance. Another approach corresponds to using lattice regression models proposed by Garcia & Gupta (2009); You et al. (2017). In this case, models are given by interpolations in a grid defined by training data. Such a class of models can be made monotonic via the choice of the interpolation strategy and recently introduced variations (Bakst et al., 2021) scale efficiently with the dimension of the input space, but downstream applications might still require different classes of models to satisfy this type of property. For neural networks, approaches such as (Nguyen & Martínez, 2019) reparameterize fully connected layers such that the gradients with respect to parameters can only be non-negative. Wehenkel & Louppe (2019), on the other hand,

---

[1]Monotonically non-increasing $f$ can be defined analogously.

consider the class of predictors $H : \mathcal{X} \mapsto \mathcal{Y}$ of the form $H(x) = \int_0^x h(t)dt + H(0)$, where $h(t)$ is a strictly positive mapping parameterized by a neural network. While such approaches guarantee monotonicity by design, they can be too restrictive or give overly complicated learning procedures. For example, the approach in (Wehenkel & Louppe, 2019) requires backpropagating through the integral. An alternative approach is based on searching over general classes of models while assigning higher importance to predictors observed to be monotonic. Similar to the case of adversarial training (Goodfellow et al., 2014), Sivaraman et al. (2020) proposed an approach to find counterexamples, i.e., pairs of points where the monotonicity constraint is violated, which are included in the training data to enforce monotonicity conditions in the next iterations of the model. However, this approach only supports fully-connected ReLU networks. Moreover, the procedure for finding the counterexamples is costly. Alternatively, Liu et al. (2020); Gupta et al. (2019) introduced point-wise regularization penalties for enforcing monotonicity, where the penalties are estimated via sampling. While Liu et al. (2020) use uniform random draws, Gupta et al. (2019) apply the regularization penalty over the training instances. Both approaches have shortcomings that we seek to address.

## 3   Monotonicity as a requirement

Given the standard supervised learning setting where $\ell : \mathcal{Y}^2 \mapsto \mathbb{R}^+$ is a loss function indicating the goodness of the predictions relative to ground truth targets, the goal is to find a predictor $h \in \mathcal{H}$ such that its expected loss – or the so-called *risk* – over the input space is minimized. Such an approach yields the empirical risk minimization framework once a finite sample is used to estimate the risk. However, given the extra monotonicity requirement, we consider an augmented framework where such a property is further enforced. We seek the optimal monotonic predictors relative to $M$, $h_M^*$:

$$h_M^* \in \arg\min_{h \in \mathcal{H}} \mathbb{E}_{x,y \sim \mathcal{X} \times \mathcal{Y}}[\ell(h(x), y)] + \gamma \Omega(h, M), \tag{1}$$

where $\gamma$ is a hyperparameter weighing the importance of the penalty $\Omega(h, M)$ which, in turn, is a measure of *how monotonic* the predictor $h$ is relative to the dimensions indicated by $M$. $\Omega(h, M)$ can be defined by the following gradient penalty (Gupta et al., 2019; Liu et al., 2020):

$$\Omega(h, M) = \mathbb{E}_{x \sim \mathcal{D}} \left[ \sum_{i \in M} \max \left( 0, -\frac{\partial h(x)}{\partial x_i} \right)^2 \right], \tag{2}$$

where $\frac{\partial h(x)}{\partial x_i}$ indicates the gradients of $h$ relative to the input dimensions $i \in M$, which are constrained to be non-negative, rendering $h$ monotonically non-decreasing relative to $M$. At this point, the only missing ingredient to define algorithms to estimate $h_M^*$ is how to define the distribution $\mathcal{D}$ over which the expectation in Eq. 2 is computed, discussed in the following sections.

### 3.1   Choosing distributions over which to compute the monotonicity penalty

In the following, we present and discuss two past choices for $\mathcal{D}$:

1) *Define $\mathcal{D}$ as the empirical distribution of the training sample*: In (Gupta et al., 2019), given a training dataset of size $N$, in addition to using the observed data to estimate the risk, the same data is used to compute the monotonicity penalty so that:

$$\Omega_{train}(h, M) = \frac{1}{N} \sum_{k=1}^N \sum_{i \in M} \max \left( 0, -\frac{\partial h(x^k)}{\partial x_i^k} \right)^2,$$

where $x^k$ indicates the $k$-th instance within the training sample. While this choice seems natural and can be easily implemented, it only enforces monotonicity in the region where the training samples lie, which can be problematic. For example, in case of covariate-shift, the test data might lie in parts of the space different from that of the training data so monotonicity cannot be guaranteed. We thus argue that one needs to enforce the monotonicity property in a region larger than what is defined by the training data. In Appendix B, we conduct an evaluation under domain shift and show the issue to become more and more relevant with the increase in the dimension $d$ of the input space $\mathcal{X}$.

2) *Define* $\mathcal{D} = \text{Uniform}(\mathcal{X})$: In (Liu et al., 2020), a simple strategy is defined so that $\Omega$ is computed over the random points drawn uniformly across the entire input space $\mathcal{X}$; i.e.:

$$\Omega_{random}(h, M) = \mathbb{E}_{x \sim \text{Uniform}(\mathcal{X})} \left[ \sum_{i \in M} \max \left( 0, -\frac{\partial h(x)}{\partial x_i} \right)^2 \right].$$

Despite its simplicity and ease of use, this approach has some flaws. In high-dimensional spaces, random draws from any distribution of bounded variance will likely lie in the boundaries of the space, hence far from the regions where data actually lie. Moreover, it is commonly observed that naturally occurring high-dimensional data is structured in lower-dimensional manifolds (c.f. (Fefferman et al., 2016) for an in-depth discussion on the manifold hypothesis). It is thus likely that random draws from the uniform distribution will lie nowhere near regions of space where training/testing data will be observed. We further illustrate the issue with examples in Appendix A, which can be summarized as follows: consider the cases of uniform distributions over the unit $n$-sphere. In such a case, the probability of a random draw lying closer to the sphere's surface than to its center is $P(\|x\|_2 > \frac{1}{2}) = \frac{2^n - 1}{2^n}$, as given by the volume ratio of the two regions of interest. Note that $P(\|x\|_2 > \frac{1}{2}) \to 1$ as $n \to \infty$, which suggests the approach in (Liu et al., 2020) will only enforce monotonicity at the boundaries of the space.

In summary, the previous approaches are either too focused on enforcing monotonicity where the training data lie, or too loose such that the monotonicity property is uniformly enforced across a large space, and the actual data manifold may be neglected. We thus propose an alternative approach where we can have some control over the volume of the input space where the monotonicity property will be enforced. Our approach uses the idea of data mixup (Zhang et al., 2018; Verma et al., 2019; Chuang & Mroueh, 2021), where auxiliary data is created via interpolations of pairs of data points, to populate areas of the space that are otherwise disregarded. Mixup was introduced by Zhang et al. (2018) with the goal of training classifiers with smooth outputs across trajectories in the input space from instances of different classes. Given a pair of data points $(x', y')$, $(x'', y'')$, the method augments the training data using interpolations given by $(\lambda x' + (1 - \lambda)x'', \lambda y' + (1 - \lambda)y'')$, where $\lambda \sim \text{Uniform}([0, 1])$. We propose using mixup to generate points where the monotonicity penalty $\Omega$ can be computed and highlight the following motivations for doing so: (1) *Interpolation* of data points more densely populates the convex hull of the training data. (2) *Extrapolation* cases where mixup is performed between data points and instances obtained at random results in points that lie anywhere between the data manifold and the boundaries of the space. We thus claim that performing mixup enables the computation of $\Omega$ on parts of the space that are disregarded if one focus only on either observed data or random draws from uninformed choices of distributions such as the uniform.

## 3.2 EVALUATION

In order to evaluate the effect of different choices of $\Omega$, we report results on three commonly used datasets covering classification and regression settings with input spaces of different dimensions. Namely, we report results for the following datasets: *Compas*, *Loan Lending Club*, and *Blog Feedback*. Models are implemented using the same architecture as in (Liu et al., 2020). Further details on the data, models, and training settings can be found in Appendix C. For all evaluation cases, we consider the baseline where training is carried out without any monotonicity enforcing penalty. For the regularized cases, the different approaches used for computing $\Omega$ are as follows: (1) $\Omega_{random}$ (Liu et al., 2020) which uses random points drawn from $\text{Uniform}(\mathcal{X})$. In this case, the sample observed at each training iteration is set to a size of 1024 throughout all experiments. (2) $\Omega_{train}$ (Gupta et al., 2019) which uses the actual data observed at each training iteration; i.e., the observed mini-batch itself is used to compute $\Omega$. And (3) $\Omega_{mixup}$ (*ours*), in which case the penalty is computed on points generated by mixing-up points from the training data and random points. In details, for each mini-batch of size $N > 1$, we augment it with complementary random data and obtain a final mini-batch of size $2N$. Out of the $\frac{2N(2N-1)}{2}$ possible pairs of points, we take a random subsample of 1024 pairs to compute mixtures of instances. In this case, we use $\lambda \sim \text{Uniform}([0,1])$ and $\lambda$ is independently drawn for each pair of points.

Results are reported in terms of both prediction performance and level of monotonicity. The latter is assessed via the fraction $\rho$ of points within a sample where the monotonicity constraint is violated; i.e., given a set of $N$ data points, we compute: $\rho = \frac{\sum_{k=1}^{N} \mathbb{1}[\min_{i \in M} \frac{\partial h(x)}{\partial x_i^k} < 0]}{N}$, such that $\rho = 0$

| | Non-mon. | $\Omega_{random}$ | $\Omega_{train}$ | $\Omega_{mixup}$ |
|---|---|---|---|---|
| | | *COMPAS* | | |
| Validation accuracy | 69.1%±0.2% | 68.5%±0.1% | 68.5%±0.1% | 68.4%±0.1% |
| Test accuracy | 68.5%±0.2% | 68.1%±0.2% | 68.0%±0.2% | 68.3%±0.2% |
| $\rho_{random}$ | 55.45%±12.26% | 0.01%±0.01% | 6.41%±4.54% | 0.00%±0.00% |
| $\rho_{train}$ | 92.98%±2.70% | 2.08%±2.21% | 0.00%±0.00% | 0.00%±0.00% |
| $\rho_{test}$ | 92.84%±2.75% | 2.16%±2.35% | 0.00%±0.00% | 0.00%±0.00% |
| | | *Loan Lending Club* | | |
| Validation RMSE | 0.213±0.000 | 0.223±0.002 | 0.222±0.002 | 0.235±0.001 |
| Test RMSE | 0.221±0.001 | 0.230±0.001 | 0.229±0.002 | 0.228±0.001 |
| $\rho_{random}$ | 99.11%±1.70% | 0.00%±0.00% | 14.47%±7.55% | 0.00%±0.00% |
| $\rho_{train}$ | 100.00%±0.00% | 7.23%±7.76% | 0.01%±0.01% | 0.00%±0.00% |
| $\rho_{test}$ | 100.00%±0.00% | 6.94%±7.43% | 0.04%±0.03% | 0.00%±0.00% |
| | | *Blog feedback* | | |
| Validation RMSE | 0.174±0.000 | 0.175±0.001 | 0.177±0.000 | 0.168±0.000 |
| Test RMSE | 0.139±0.001 | 0.139±0.001 | 0.142±0.001 | 0.143±0.001 |
| $\rho_{random}$ | 76.17%±12.37% | 0.05%±0.08% | 3.86%±4.19% | 0.00%±0.01% |
| $\rho_{train}$ | 78.67%±5.28% | 78.59%±6.37% | 0.01%±0.01% | 0.01%±0.01% |
| $\rho_{test}$ | 76.29%±6.47% | 78.99%±7.20% | 0.02%±0.02% | 0.02%±0.02% |

Table 1: Evaluation results in terms of 95% confidence intervals resulting from 20 independent training runs. Results correspond to the checkpoint that obtained the best prediction performance on validation data throughout training. The lower the values of $\rho$ the better.

corresponds to monotonic models over the considered points. Moreover, in order to quantify the degree of monotonicity in different parts of the space, we compute $\rho$ for 3 different sets of points: (1) $\rho_{random}$, computed on a sample drawn according to Uniform($\mathcal{X}$). We used a sample of 10,000 points throughout the experiments. (2) $\rho_{train}$, computed on the training data. And (3) $\rho_{test}$: computed on the test data. Results are summarized in Table 1 in terms of both prediction performance along with the metric indicating the *degree of monotonicity* of the predictor for each regularization strategy. Prediction performance is measured in terms of accuracy for classification tasks, and RMSE for the case of regression. Monotonicity, on the other hand, is quantified via the fraction $\rho$ of points within a sample where the monotonicity constraint is violated. Results reported in the tables represent 95% confidence intervals corresponding to multiple independent training runs. Across all datasets, the different penalties do not result in significant variations in the performance of the final predictors. This indicates that the class of predictors corresponding to the subset of $\mathcal{H}$ that is monotonic relative to $M$, denoted $\mathcal{H}_M$, has enough capacity so as to be able to match the performance of the best canditates within $\mathcal{H}$. In terms of monotonicity, we observe a clear pattern leading to the following intuition: *monotonicity is achieved in the regions where it is enforced.* This is evidenced by the observation that $\rho_{random}$ is consistently lower for $\Omega_{random}$ relative to $\Omega_{train}$ and $\Omega_{mixup}$ while, on the other hand, $\rho_{train}$ and $\rho_{test}$ are consistently lower for $\Omega_{train}$ and $\Omega_{mixup}$ compared to $\Omega_{random}$. A comparison between $\Omega_{train}$ and $\Omega_{mixup}$ shows what we anticipated: enforcing monotonicity in points resulting from mixup yields predictors that are as monotonic as those given by the use of $\Omega_{train}$ in actual data, but significantly better at the boundaries of $\mathcal{X}$. Finally, the results demonstrate that our proposed approach $\Omega_{mixup}$ achieves the best results in terms of monotonicity for all the sets of points that we considered. Moreover, our approach introduces no significant computation overhead. Algorithm 1 in Appendix C details on how to compute $\Omega_{mixup}$.

## 4 MONOTONICITY AS A REGULARIZER

In Section 3, we presented an efficient approach to enforce monotonicity when it is a requirement. We now consider a different perspective and show that adding monotonicity constraints during training can yield extra benefits to trained models. In these cases monotonicity is not a requirement, and hence it is not necessary for it to be satisfied everywhere. As such, the penalties we discuss from now on are computed considering only data points, and no random draws are utilized. In the following sections, we introduce notions of monotonicity that will be enforced in our models, and discuss advantages of using monotonicity for different applications such as for the detection of anomalous data and controllable generative modelling. In Appendix F, we consider a further application for cases where one's interest is to obtain explanations from observed predictions.

## 4.1 GROUP MONOTONICITY

First, we consider the case of $K$-way classifiers realized through convolutional neural networks. In this case, data examples correspond to pairs $x, y \sim \mathcal{X} \times \mathcal{Y}$, but $\mathcal{Y} = \{1, 2, 3, ..., K\}$, $K \in \mathbb{N}$. Models parameterize a data-conditional categorical distribution over $\mathcal{Y}$, i.e., for a given model $h$, $h(x)_y$ will yield likelihoods for each class indexed in $\mathcal{Y}$. Under this setting, we introduce the notion of *Group Monotonicity*: we aim to find the models $h$ such that the outputs corresponding to each class satisfy a monotonic relationship with a specific subset of high-level representations given by some inner convolutional layer. Let the outputs of a specific layer within a convolutional model be represented by $a_w$, $w \in [1, 2, 3, ..., W]$, where $W$ indicates the width of the chosen layer given by its number of output feature maps. For simplicity of exposition, we consider the rather common case of convolutional layers where each feature map $a_w$ is 2-dimensional. We then partition such a set of representations into disjoint subsets, or *slices*, of uniform sizes. Each subset is then paired with a particular output or class, and hence denoted by $S_k$, $k \in \mathcal{Y}$. An illustration is provided in Figure 1, where a generic convolutional model has the outputs of a specific layer partitioned into slices $S_k$ which are paired with the corresponding output units.

**Definition 2** *Group monotonic convolutional classifiers:* We say $h$ is group monotonic for input $x$ and its corresponding class label $y$ if $h(x)_y$ is partially monotonic relative to all elements in $S_y$.

We highlight that in this case, unlike the discussion in Section 3, monotonicity *is not* an application requirement, and it does not need to be satisfied everywhere. Intuitively, our goal is to "reserve" groups of high-level features to be monotonic with respect to different classes. Such an structure can add extra benefits to models, e.g., more accurate anomaly detection. For training, we perform monotonic risk minimization as described in Eq. 1, and the risk is given by the negative log-likelihood over training points. Moreover, we design a penalty $\Omega$ that focuses only on observed data points during training and penalizes the slices of the Jacobian corresponding to a given class, i.e., a cross-entropy criterion enforces larger gradients on the class slice. In order to formally introduce such a penalty, denoted by $\Omega_{group}$, we first define the total gradient $O_k$, $k \in \mathcal{Y}$, of a slice $S_k$ as follows: $O_y(x) = \sum_{a_w \in S_y} \sum_{i,j} \frac{\partial h(x)_y}{\partial a_{w,i,j}}$, where the inner sum accounts for spatial dimensions of $a_w$. Given the set of total gradients, a batch of size $m$, and inverse temperature $\mu$, $\Omega_{group}$ will be:

$$\Omega_{group} = -\frac{1}{m} \sum_{i=1}^{m} \log \frac{e^{\frac{O_{y^i}(x^i)}{\mu}}}{\sum_{k=1}^{K} e^{\frac{O_k(x^i)}{\mu}}}. \tag{3}$$

## 4.2 EVALUATION

### 4.2.1 ASSESSING PERFORMANCE OF GROUP MONOTONIC CLASSIFIERS

We start our evaluation by verifying whether the group monotonicity property can be effectively enforced into classifiers trained on standard object recognition benchmarks. In order to do so, we verify the performance of the *total activation classifier*, as defined by: $\arg\max_{k \in \mathcal{Y}} T_k(x)$, where $T_k$ indicates the total activation on slice $S_k$: $T_k(x) = \sum_{a_w \in S_k} \sum_{i,j} a_{w,i,j}(x)$. A good prediction performance of such a classifier serves as evidence that the group monotonicity property is satisfied by the model over the test data under consideration since it indicates the slice relative to the underlying class of test instances has the highest total activation. We thus run evaluations for both CIFAR-10 and ImageNet, and classifiers in each case correspond to WideResNets (Zagoruyko & Komodakis, 2016) and ResNet-50 (He et al., 2016), respectively. Training details are presented in Appendix D.

Results are reported in Table 2 and compared based on the top-1 prediction accuracy measured on the test data. We use standard classifiers as the baselines where no monotonicity penalty is applied in order to isolate the effect of the penalty. In both datasets, the total activation classifiers for group monotonic models (indicated by the prefix *mono*) are able to approximate the performance of

| Model | $\arg\max_{k \in \mathcal{Y}} h(x)_k$ | $\arg\max_{k \in \mathcal{Y}} T_k(x)$ |
|---|---|---|
| CIFAR-10 | | |
| WideResNet | 95.46% | 16.35% |
| *Mono*WideResNet | 95.64% | 94.95% |
| ImageNet | | |
| ResNet-50 | 75.85% | 0.10% |
| *Mono*ResNet-50 | 76.50% | 72.52% |

Table 2: Top-1 accuracy of standard and group monotonic models.

the classifier defined at the output layer, $\arg\max_{k \in \mathcal{Y}} h(x)_k$. This suggests that the higher total activation generally matches the predicted class for group monotonic models, which indicates the property is successfully enforced. Considering performances obtained at the output layer, there were small variations in accuracy when we included monotonicity penalties, which should be considered in practical uses of group monotonicity. Nonetheless, results suggest that one can perform closely to unconstrained models while focusing on the set of group monotonic candidates. Additional experiments are reported on Table 8 on Appendix E for cases with small sample sizes, where we show that the performance of the classifier defined at the output layer upper bounds that of the total activation classifier, i.e., *the better the underlying classifier the more group monotonic it can be made*.

### 4.2.2 USING GROUP MONOTONICITY TO DETECT ANOMALIES

After showing that group monotonicity can be enforced successfully without affecting the prediction performance, we discuss approaches to leverage it and introduce applications of the models satisfying such a property. In particular, we consider the application of detecting anomalous instances, i.e., those where the model may have made a mistake. For example, consider the case where a classifier is deployed to production and, due to some problem external to the model, it is queried to do prediction for an input consisting of white noise. Standard classifiers would provide a prediction even for such a clearly anomalous input. However, a more desirable behavior is to somehow indicate that the instance is problematic. We claim that imposing structure in the features, e.g., by enforcing group monotonicity, can help in deciding when not to predict. To evaluate the proposed method, we implement anomalous test instances using adversarial perturbations. Namely, we create $L_\infty$ PGD attackers (Madry et al., 2017) and detect anomalies based on simple statistics of the features. In details, for a given input $x$, we compute the normalized entropy $H^*(x)$ of the categorical distribution defined by the application of the softmax operator over the set of total activations $T_\mathcal{Y}(x)$: $H^*(x) = \frac{\sum_{k \in \mathcal{Y}} p_k(x) \log p_k}{\log K}$, where $K = |\mathcal{Y}|$ and the set $p_\mathcal{Y}(x)$ corresponds to the parameters of a categorical distribution defined by: $p_\mathcal{Y}(x) = \text{softmax}(T_\mathcal{Y}(x))$. Decisions can then be made by comparing $H^*(x)$ with a threshold $\tau \in [0, 1]$, defining the detector $\mathbb{1}_{\{H^* > \tau\}}$. We evaluate the detection performance of this approach on both MNIST and CIFAR-10. Training for the case of CIFAR-10 follows the same setup discussed on Section 4.2.1. For MNIST on the other hand, we modify the standard LeNet architecture by increasing the width of the second convolutional layer from 64 to 150. This layer is then used to enforce the group monotonicity property. The resulting model is referred to as WideLeNet. Moreover, $\gamma$ and $\mu$ are set to $1e10$ and $1$, respectively. Adversarial attacks are created under the white-box setting, i.e., by exposing the full model to the attacker. The perturbation budget in terms of $L_\infty$ distance is set to $0.3$ and $\frac{8}{255}$ for the cases of MNIST and CIFAR-10, respectively. Detection performance is reported in Table 3 for the considered cases in terms of the area under the operating curve (AUC-ROC). The baselines are the models for which the monotonicity penalty is not enforced. They are trained under the same conditions and the same computation budget as the models where the penalty is enforced. The results are as expected, i.e., for monotonic models, test examples for which the total activations are not structured very often correspond to anomalous inputs.

Finally, due to space constraints, we discuss the application of group monotonicity to explainability in appendix F.

| Model | AUC-ROC |
|---|---|
| MNIST | |
| WideLeNet | 54.47% |
| *Mono*WideLeNet | 100.00% |
| CIFAR-10 | |
| WideResNet | 67.35% |
| *Mono*WideResNet | 79.33% |

Table 3: AUC-ROC (the higher the better) for the detection of adversarially perturbed data instances.

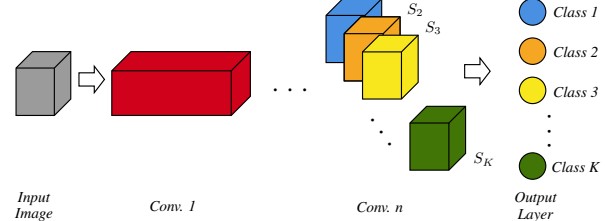

Figure 1: Illustration of a group monotonic convolutional model. Representations output by a certain convolutional layer are partitioned into disjoint subsets.

### 4.3 Disentangled representation learning under monotonicity constraints

We now discuss another application where using monotonicity as a regularizer can be beneficial. In particular, we consider the case of learning a disentangled set of representations. In this case, it is often assumed that the latent variables are independent, and hence control over generative factors can be achieved, e.g., one can modify a specific aspect of the data by modifying the value of a corresponding latent variable. However, we argue that disentanglement is necessary but not sufficient to enable controllable data generation. We need latent variables that satisfy some notion of monotonicity to be able to decide their values resulting in desired properties. For example, assume we are interested in generating images of simple geometric forms and desire to control factors such as shape and size. In this example, even if a disentangled set of latent variables is available, we cannot decide how to change the value of the latent variable to get a bigger or a smaller object if there is no monotonic relationship between the size and the value of the corresponding latent variable. We address this issue and build upon the weakly supervised framework introduced by Locatello et al. (2020). This work extends the popular $\beta$-VAE setting (Higgins et al., 2016) by introducing weak supervision such that the training instances are presented to the model in pairs $(x^1, x^2)$ where only one or a few generative factors are changing between each pair. Here, we propose to apply group monotonocity over the activations of the corresponding latent variables to have more controlable factors. In the VAE setting, the data is generated according to $p(x|z)p(z)$ given the latent variables $z$. Approximation is then performed by introducing $p_\theta(x|z)$ and $q_\phi(z|x)$, both parameterized by neural networks. Our goal is to have $z$ fully factorizable in its dimensions, i.e., $p(z) = \prod_{i=1}^{Dim[z]} p(z_i)$, which needs to be captured by the approximate posterior distribution $q_\phi(z|x)$. Training is performed by maximization of a lower-bound on the likelihood of the data, which yields the following loss:

$$\mathcal{L}_{ELBO} = \mathbb{E}_{x^1, x^2} \sum_{i \in \{1,2\}} -\mathbb{E}_{\tilde{q}_\phi(\hat{z}|x^i)} \log(p_\theta(x^i|\hat{z})) + \beta D_{KL}(\tilde{q}_\phi(\hat{z}|x^i), p(\hat{z})), \qquad (4)$$

where $\tilde{q}_\phi(\hat{z}_j|x^i) = q_\phi(z_j|x^i)$ for the latent dimensions $z_i$ that change across $x^1$ and $x^2$, and $\tilde{q}_\phi(\hat{z}_j|x^i) = \frac{1}{2}(q_\phi(\hat{z}_j|x^1) + q_\phi(\hat{z}_j|x^2))$ for those that are common (i.e., the approximate posterior of the shared latent variables are forced to be the same for $x^1$ and $x^2$). The outer expectation is estimated by sampling pairs of data instances $(x^1, x^2)$ where only a number of generative factors vary. In our experiments, we consider the case where exactly one generative factor changes across inputs. Moreover, we follow Locatello et al. (2020) and assign the changing factor, denoted by $y$, to the dimension $j$ of $z$ such that $y = \arg\max_{j \in Dim[z]} D_{KL}(z_j^1, z_j^2)$. While the above objective enforces disentanglement, controllable generation requires some regularity in $z$ so that users can decide values of $z$ resulting in desired properties in the generated samples. We introduce $\Omega_{VAE}$ to enforce such a regularity, i.e., a monotonic relationship is enforced for the distance between pairs of images where only a particular generative factor vary and a corresponding latent variable. In other words, an increasing trend in the value of each dimension of $z$ should yield a greater change in the output along a generative factor. Formally, $\Omega_{VAE}$ is defined as follows:

$$\Omega_{VAE} = -\frac{1}{2m} \sum_{i=1}^{m} \log \frac{e^{\frac{L(x^{i,1}, x^{i,2}, y^i)}{\mu}}}{\sum_{k=1}^{K} e^{\frac{L(x^{i,1}, x^{i,2}, k)}{\mu}}} + \log \frac{e^{\frac{L(x^{i,2}, x^{i,1}, y^i)}{\mu}}}{\sum_{k=1}^{K} e^{\frac{L(x^{i,2}, x^{i,1}, k)}{\mu}}}, \qquad (5)$$

where $L$ is given by the gradient of the mean squared error (MSE) between pairs of images that are 1-factor away with respect to the dimension $y$ of $z$ assigned to the changing factor, i.e.:

$$L(x^i, x^j, y) = \frac{\partial \text{MSE}(\hat{x}^i, x^j)}{\partial \tilde{z}_y}. \qquad (6)$$

In this case, $\hat{x}$ indicates the reconstruction of $x$. We evaluate such an approach by training the same 4-layered convolutional VAEs described in (Higgins et al., 2016) using the 3d-shapes dataset[2]. The dataset is composed of images containing shapes generated from 6 independent generative factors: *floor color*, *wall color*, *object color*, *scale*, *shape* and *orientation*. All combinations of these factors are present exactly once, resulting in $m = 480000$. We compared VAEs trained with and without the inclusion of the monotonicity penalty given by $\Omega_{VAE}$. We highlight that the goal of

---

[2]https://github.com/deepmind/3d-shapes

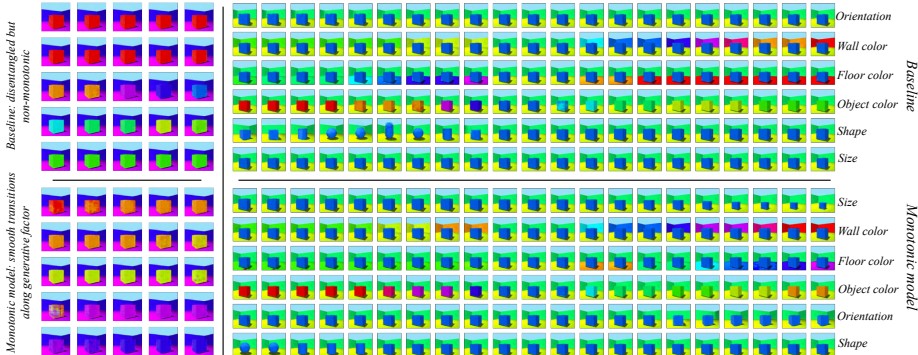

Figure 2: Comparisons between data generated by standard and monotonic models. On the two panels on the left, we compare generations from a linear combination of the latent code of 2 images which only differs in the object color. On the two panels vertically stacked on the right, we start from the same image but change one latent dimension at a time.

the proposed framework is not to improve over current approaches in terms of *how disentangled* the learned representations are. Rather, we seek to achieve similar results in that sense, but impose extra regularity and structure in the relationship between the generated images and the values of $z$ so that the generative process is more easily *controllable*. Qualitative analysis is performed and shown in Figure 2. The two panels on the left represent the data generated by a linear combination of the latent code corresponding to two images that only vary in the factor *object color*. The panels stacked on the right present a per-dimension traversal of the latent space starting from a common image. It can be observed that disentanglement is indeed achieved in both cases. The monotonic model presents much smoother transitions between colors while the base model gives long sequences of very close images followed by very sharp transitions where the colors sometimes repeat (e.g., green-yellow-green transitions in the fourth row). As for the results per factor, the monotonic model provides more structure in the latent space compared to the base model. This can be observed in the shape factor. The monotonic model provides a certain order: sphere, cylinder, and then cube. Visually inspecting many samples, the monotonic model is following this order for the generated shapes. This pattern is even more pronounced in the color factors. We have found that the colors generated by the monotonic model follows the order of the colours in *the HUE cycle*. So our model has ordered the latent space and we know how to navigate it to generate a desired image. On the other hand, the baseline has no clear order of the latent space. For example, the baseline generates cubes at different ranges of $z$. Similarly, the colors generated by the baseline model do not have a clear order. To further support the claim that $\Omega_{VAE}$ induces regularity in the latent space, we introduce the analysis shown in Table 4. We started by increasing $z_3$ (associated to *floor color* for both models), and recorded the sequence of the generated colors. We observed that for a large fraction of the data, the monotonic models yield sequences of images where the color of the floor is ordered according to its corresponding HUE angle. Further details are available in Appendix H.

| Model | HUE structured rate |
|---|---|
| Base model | 0.00% |
| Mon. model | 89.44% |

Table 4: rate of examples where colors are sorted according to hue.

## 5 CONCLUSION

We proposed approaches that enable learning algorithms based on risk minimization to find solutions that satisfy some notion of monotonicity. First, we discussed the case where monotonicity is a *design requirement* that needs to be satisfied everywhere. In this case, we identified limitations in prior work that resulted in models satisfying the property only in very specific parts of the space. We then introduced an efficient procedure that was observed to significantly improve the solutions in terms of the volume of the space where the monotonicity requirement is achieved. In addition, we further argued that, even when not required, *models satisfying monotonicity present useful properties*. We studied the case of image classifiers and generative models and showed that imposing structure in learned representations via group monotonicity is beneficial and can be done efficiently.

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

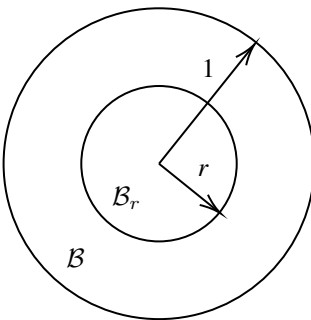

Figure 3: Illustration unit spheres $\mathcal{B}$ and $\mathcal{B}_r$ on the plane.

# A    ILLUSTRATIVE EXAMPLES ON THE SPHERE: MIXUP HELPS TO POPULATE THE SMALL VOLUME INTERIOR REGION

To further illustrate the issue discussed in the item 2 of Section 3.1 as well the effect of our proposal, we discuss a simple example considering random draws from the unit $n$-sphere, shown in Figure 3, i.e., the set of points $\mathcal{B} = \{x \in \mathbb{R}^n : ||x||_2 < 1\}$. We further consider a concentric sphere of radius $0 < r < 1$ given by $\mathcal{B}_r = \{x \in \mathbb{R}^n : ||x||_2 < r\}$. We are interested in the probability of a random draw from $\mathcal{B}$ to lie outside of $\mathcal{B}_r$, i.e.: $P(||x||_2 > r), x \sim \mathcal{D}(\mathcal{B})$, for some distribution $\mathcal{D}$. We start by defining $\mathcal{D}$ as the Uniform$(\mathcal{B})$, which results in $P(||x||_2 > r) = 1 - r^n$. In Figure 4a, we can see that for growing $n$, $P(||x||_2 > r)$ is very large even if $r \approx 1$, which suggests most random draws will lie close to $\mathcal{B}$'s boundary.

We now evaluate the case where mixup is applied and random draws are taken in two steps: we first observe $y \sim \text{Uniform}(\mathcal{B})$, and then we perform mixup between $y$ and the origin[3], i.e., $x = \lambda y$, $\lambda \sim \text{Uniform}([0, 1])$. In this case, $P(||x||_2 > r) = (1 - r^n)(1 - r)$, which is shown in Figure 4b as a function of $r$ for increasing $n$. We can then observe that even for large $n$, $P(||x||_2 > r)$ decays linearly with $r$, i.e., we populate the interior of $\mathcal{B}$ and $x$ in this case follows a non-uniform distribution such that its norms histogram is uniform.

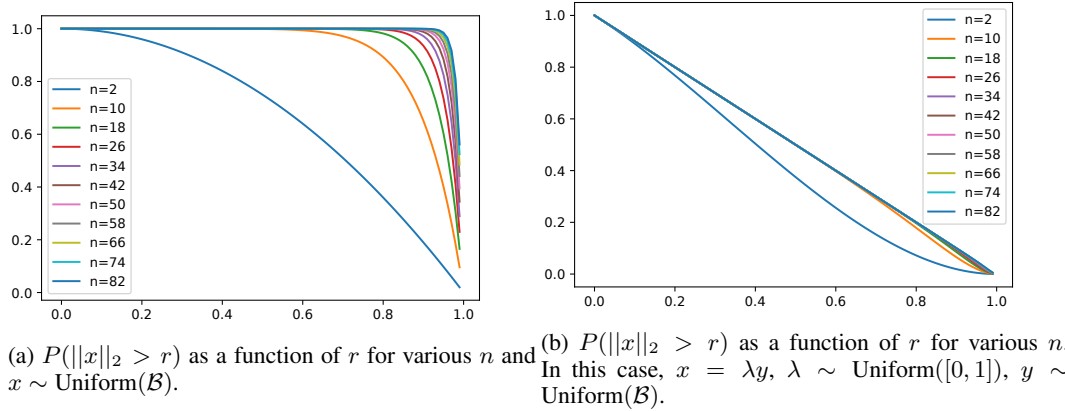

(a) $P(||x||_2 > r)$ as a function of $r$ for various $n$ and $x \sim \text{Uniform}(\mathcal{B})$.

(b) $P(||x||_2 > r)$ as a function of $r$ for various $n$. In this case, $x = \lambda y$, $\lambda \sim \text{Uniform}([0, 1])$, $y \sim \text{Uniform}(\mathcal{B})$.

Figure 4: Illustrative example showing that uniformly distributed draws on a unit sphere in $\mathbb{R}^n$ concentrate on its boundary for large $n$. Applying mixup populates the interior of the space.

---

[3]Similar conclusions hold for any fixed point within $\mathcal{B}$. The origin is chosen for convenience.

# B  PROOF-OF-CONCEPT EVALUATION

We start by describing the approach we employ to generate data containing the properties required by our evaluation. Denote a design matrix by $X_{N \times D}$ such that each of its $N$ rows corresponds to a feature vector within $\mathbb{R}^D$. In order to ensure the data lies in some manifold, we first obtain a low-dimensional synthetic design matrix given by $X'_{N \times d}$, where each entry is sampled randomly from Uniform($[-10, 10]$). We then expand it to $\mathbb{R}^D$ by applying the following transformation:

$$X = X'A, \tag{7}$$

where the expansion matrix given by $A_{d \times D}$ is such that each of its entries are independently drawn from Uniform($[0, 1]$). Throughout our experiments, $d = \lfloor 0.3D \rfloor$ was employed.

Target values for the function $f$ to be approximated are defined as sums of functions of scalar arguments applied independently over each dimension. We thus select a set of dimensions $M \in [D]$ with respect to which $f$ is to be monotonic, i.e.:

$$f(x) = \sum_{i \in M} g_i(x_i) + \sum_{j \in \bar{M}} h_j(x_j), \tag{8}$$

and every $g_i : \mathbb{R} \mapsto \mathbb{R}$ is increasing monotonic, while every $h_i : \mathbb{R} \mapsto \mathbb{R}$ is not monotonic.

We then create two evaluation datasets. One of them, referred to as the validation set, is identically distributed with respect to $X$ since it is obtained following the same procedure discussed above. In order to simulate covariate-shift, we create a test set by changing the expansion matrix $A$ to a different one.

$$X_{val} = X'_{val}A, \quad X_{test} = X'_{test}A_{test}, \tag{9}$$

where $A_{test}$ will be given by entry-wise linear interpolations between $A$, used to generate the training data, and a newly sampled expansion matrix $A'$: $A_{test} = \alpha A' + (1 - \alpha)A$. The parameter $\alpha \in [0, 1]$, set to $0.8$ in the reported evaluation, controls the shift between $A_{test}$ and $A_{test}$ in terms of the Frobenius norm, which in turn enables the control of how much the test set shifts relative to the training data.

We thus trained models to approximate $f$ for spaces of increasing dimensions as well as for an increasing number of dimensions with respect to which $f$ is monotonic. Results are reported in Table 5 in terms of RMSE on the two evaluation datasets, and in terms of monotonicity in Table 6 where $\rho$ is computed both on random points and on the shifted test set. Entries in the tables correspond to the centers of 95% confidence intervals resulting from 20 independent training runs.

We highlight the two following observations regarding the prediction performances shown in table 5: different models present consistent performances across evaluations, which suggests different monotonicity-enforcing penalties do not significantly affect prediction accuracy. Moreover, the proposed approach used to generate test data under covariate-shift is effective given the gap in performance consistently observed between the validation and the test partitions. In terms of monotonicity, results in Table 6 suggest that $\Omega_{random}$ and $\Omega_{train}$ are only effective on either random or data points, which seems to aggravate when the dimension $D$ grows. $\Omega_{mixup}$, on the other hand, is effective on both sets of points, and continues to work well for growing $D$. Furthermore, covariate-shift significantly affects $\Omega_{train}$ for higher-dimensional cases, while $\Omega_{mixup}$ performs well in such a case.

| $|M|/D$ | 20/100 | | 40/200 | | 80/400 | | 100/500 | |
|---|---|---|---|---|---|---|---|---|
| | Valid. RMSE | Test RMSE | Valid. RMSE | Test RMSE | Valid. RMSE | Test RMSE | Valid. RMSE | Test RMSE |
| Non-mon. | 0.007 | 0.107 | 0.006 | 0.082 | 0.007 | 0.087 | 0.011 | 0.146 |
| $\Omega_{random}$ | 0.008 | 0.117 | 0.006 | 0.081 | 0.007 | 0.093 | 0.012 | 0.125 |
| $\Omega_{train}$ | 0.008 | 0.115 | 0.006 | 0.086 | 0.007 | 0.089 | 0.012 | 0.134 |
| $\Omega_{mixup}$ | 0.008 | 0.114 | 0.007 | 0.084 | 0.008 | 0.088 | 0.012 | 0.134 |

Table 5: Prediction performance of models trained on generated data in spaces of growing dimension ($D$) and number of monotonic dimensions ($|M|$). Different regularization strategies do not affect prediction performance. The performance gap consistently observed across the evaluation sets highlights the shift between the two sets of points. The lower the values of RMSE the better.

| $|M|/D$ | 20/100 | | 40/200 | | 80/400 | | 100/500 | |
|---|---|---|---|---|---|---|---|---|
| | $\rho_{random}$ | $\rho_{test}$ | $\rho_{random}$ | $\rho_{test}$ | $\rho_{random}$ | $\rho_{test}$ | $\rho_{random}$ | $\rho_{test}$ |
| Non-mon. | 99.90% | 99.99% | 97.92% | 94.96% | 98.47% | 96.56% | 93.98% | 90.01% |
| $\Omega_{random}$ | 0.00% | 3.49% | 0.00% | 4.62% | 0.01% | 11.36% | 0.02% | 19.90% |
| $\Omega_{train}$ | 1.30% | 0.36% | 4.00% | 0.58% | 9.67% | 0.25% | 9.25% | 5.57% |
| $\Omega_{mixup}$ | 0.00% | 0.35% | 0.00% | 0.44% | 0.00% | 0.26% | 0.00% | 0.42% |

Table 6: Fraction of monotonic points $\rho$ for models trained on generated data in spaces of growing dimension ($D$) and number of monotonic dimensions ($|M|$). Different regularization strategies is effective on only one of $\rho_{random}$ or $\rho_{test}$, while $\Omega_{mixup}$ seems effective throughout conditions. The lower the values of $\rho$ the better.

## C  DATASETS, MODELS, AND TRAINING DETAILS FOR EXPERIMENTS REPORTED IN SECTION 3.2

Algorithm 1 describes a procedure used to compute the proposed regularization $\Omega_{mixup}$.

---

**Algorithm 1** Procedure to compute $\Omega_{mixup}$.

---

*Input* mini-batch $X_{[N \times d]}$, model $h$, monotonic dimensions $M$
$X_\Omega = \{\}$  # Initialize set of points used to compute regularizer.
$\tilde{X}_{[N \times d]} \sim \text{Uniform}(\mathcal{X}^N)$  # Sample random mini-batch with size $N$.
$\hat{X} = \text{concat}(X, \tilde{X})$  # Concatenate data and random batches.
**repeat**
    $i, j \sim \text{Uniform}(\{1, 2, ..., 2N\}^2)$  # Sample random pair of points.
    $\lambda \sim \text{Uniform}([0, 1])$
    $x = \lambda \hat{X}^i + (1 - \lambda)\hat{X}^j$  # Mix random pair.
    $X_\Omega.\text{add}(x)$  # Add $x$ to set of regularization points.
**until** Maximum number of pairs reached
$\Omega_{mixup}(h, M) = \frac{1}{|X_\Omega|} \sum_{x \in X_\Omega} \sum_{i \in M} \max\left(0, -\frac{\partial h(x)}{\partial x_i}\right)^2$
**return** $\Omega_{mixup}$

---

In Table 7, we list details on the three datasets used to evaluate our proposals as reported in Section 3.2.

| Dataset | Dim[$\mathcal{X}$] | $|M|$ | # Train | # Test | Task |
|---|---|---|---|---|---|
| *Compas*[4] | 13 | 4 | 4937 | 1235 | *Classification* |
| *Loan Lending Club*[5] | 33 | 11 | 8500 | 1500 | *Regression* |
| *Blog Feedback*[6] | 280 | 8 | 47287 | 6904 | *Regression* |

Table 7: Description of datasets used for empirical evaluation.

Models follow the architecture in (Liu et al., 2020) using dense layers whose weights are kept separate in early layers for the input dimensions with respect to which monotonicity is to be enforced. We set the depth of all networks to 3, and use a bottleneck of size 10 for two datasets (Compas and Loan Lending Club), and 100 for the case of the Blog Feedback dataset and the experiments on generated data. Training is carried out with the Adam optimizer (Kingma & Ba, 2014) with a global learning rate of $5\mathrm{e}{-3}$, and $\gamma$ is set to 1e4. The training batch size is set to 256 throughout experiments.

---

[4] https://www.kaggle.com/danofer/compass
[5] https://www.openintro.org/data/index.php?data=loans_full_schema
[6] https://archive.ics.uci.edu/ml/datasets/BlogFeedback

| Model | $\arg\max_{k\in\mathcal{Y}} h(x)_k$ | $\arg\max_{k\in\mathcal{Y}} T_k(x)$ |
|---|---|---|
| 10% | | |
| WideResNet | 85.68% | 16.35% |
| *Mono*WideResNet | 85.77% | 82.21% |
| 30% | | |
| WideResNet | 92.12% | 14.51% |
| *Mono*WideResNet | 92.42% | 88.88% |
| 60% | | |
| WideResNet | 94.51% | 10.08% |
| *Mono*WideResNet | 94.86% | 93.81% |

Table 8: Top-1 accuracy obtained by both standard and group monotonic models on sub-samples of CIFAR-10. Predicition performance obtained by classifiers defined by the total activations is upper bounded by the performance obtained at the output layer for monotonic models.

## D  MODELS AND TRAINING DETAILS FOR EXPERIMENTS REPORTED IN SECTION 4

For the case of CIFAR-10, WideResNets (Zagoruyko & Komodakis, 2016) are used. The models are initialized randomly and trained both with and without the monotonicity penalty. Standard stochastic gradient descent (SGD) implements the parameters update rule with a learning rate starting at 0.1, being decreased by a factor of 10 on epochs 10, 150, 250, and 350. Training is carried out for a total of 600 epochs with a batch size of 64. For ImageNet, on the other, training consists of fine tuning a pre-trained ResNet-50, where the fine-tuning phase included the monotonicity penalty. We do so by training the model for 30 epochs on the full ImageNet training partition. In this case, given that the label set $\mathcal{Y}$ is relatively large, using the standard ResNet-50 would result in small slices $S_k$. To avoid that, we add an extra final convolution layer with $W = 15K$. Training is once more carried out with SGD using a learning rate set to 0.001 in this case, and reduced by a factor of 5 at epoch 20. In both cases, the group monotonicity property is enforced at the last convolutional layer. Other hyperparameters such as the strength $\gamma$ of the monotonicity penalty as well as the inverse temperature $\mu$ used to compute $\Omega_{group}$ are set to 1 and 50 for the case of CIFAR-10, and to 5 and 10 for the case of ImageNet. Both momentum and weight decay are further employed and their corresponding parameters are set to 0.9 and 0.0001. For MNIST classifiers, training is performed for 20 epochs using a batch size of 64 and the Adadelta optimizer (Zeiler, 2012) with a learning rate of 1.

## E  ENFORCING GROUP MONOTONICITY UNDER SMALL SAMPLES

Using CIFAR-10, we further evaluate how the proposed group monotonicity penalty behaves in data-constrained settings, i.e., we check whether or not the property can be enforced under small sample regimes. We do so by sub-sampling the original training data by randomly selecting a fraction of the training images uniformly across classes. We then train the same WideResNet for the same computation budget in terms of number of iterations as the models trained in the complete set of images. The learning rate schedule also matches that of the training on the full dataset in that the learning rate is reduced at exactly the same iterations across all training cases. Results are reported in Table 8 for sub-samples corresponding to 10%, 30%, and 60% of CIFAR-10. Results are consistent across the three sets of results in showing that predictions obtained from the total activation of feature slices approximate the prediction performance of the underlying model for the case of group monotonic predictors, i.e., the extent to which the underlying model is able to accurately predict correct classes upper bound the resulting "level of monotonicity". In simple terms, the better the classifier, the more group monotonic it can be made.

## F  SELECTING FEATURE MAPS TO COMPUTE VISUAL EXPLANATIONS

Approaches based on Class Activation Maps (CAM) such as Grad-CAM and its variations (Selvaraju et al., 2017; Chattopadhay et al., 2018) seek to extract *explanations* from convolutional models. By

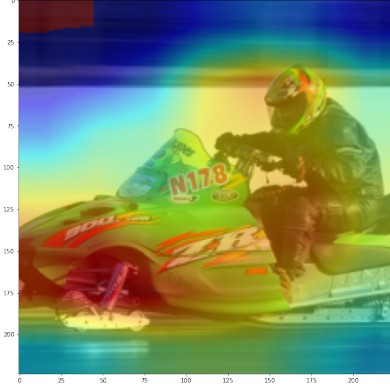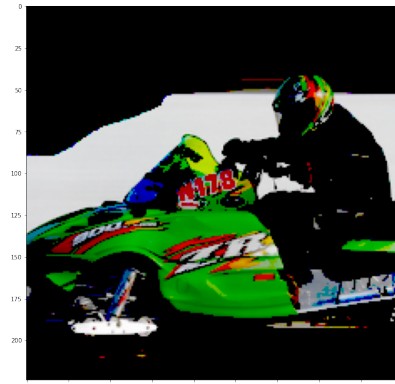

Figure 5: Example of explanation heat-map and corresponding occlusion obtained with Grad-CAM and a ResNet-50 trained on ImageNet. The example belongs to the validation set and corresponds to the class *snowmobile*.

explanation we mean to refer to indications of properties of the data implying the predictions of a given model. Under such a framework, one can obtain so-called explanation heat-maps through the following steps: (1) Compute a weighted sum of activations of feature maps in a chosen layer; (2) Upscale the results in order to match the dimensions of the input data; (3) Superimpose results onto the input data. Specifically for the case of applications to image data, following those steps results in highlighting the patches of the input that were deemed relevant to yield the observed predictions. Different approaches were then introduced in order to define the weights used in the first step. A very common choice is to use the total gradient of the output corresponding to the prediction with respect to activations of each feature map.

For the case of group monotonic classifiers, we are interested in verifying whether one can define useful explanation heat-maps by considering only the feature slices corresponding to the predicted class, i.e., for a given input pair $(x, y)$, we compute explanation heat-maps considering only its corresponding feature activation slice $S_y(x)$. We thus design an experiment to evaluate the effectiveness of such an approach by using external auxiliary classifiers to perform predictions from test data that was occluded using explanation heat-maps obtained using different models and sets of representations. In other words, we use the explanation maps to remove from the data the parts that were not indicated as relevant. We then assume that good explanation maps will be such that classifiers are able to correctly classify occluded data since relevant patches are conserved. In further details, occlusions are computed by first applying a CAM operator given a model $h$ and data $x$, which results in a heat-map with entries in $[0, 1]$. We then use such a heat-map as a multiplicative mask to get an occluded version of $x$, denoted $x'$, i.e.:

$$x' = \text{CAM}(x, h) \circ x, \tag{10}$$

where the operator $\circ$ indicates element-wise multiplication. An example of such a procedure is shown in Figure 5. We apply the above procedure to all of the validation data, and use resulting points to then assess the prediction performance of auxiliary classifiers.

Explanation maps are computed using the same models discussed in Section 4.2.1 for ImageNet. The CAM operator corresponds to a variation of Grad-CAM++ (Chattopadhay et al., 2018) where the model activations are directly employed for weighing feature maps rather than the gradients. We consider 4 auxiliary pre-trained classifiers corresponding to ResNext-50 (Xie et al., 2017), MobileNet-v3 (Howard et al., 2019), VGG-16 (Simonyan & Zisserman, 2014), and SqueezeNet (Iandola et al., 2016). Results are reported in Table 9 which also include the reference performance of the auxiliary classifiers on the standard validation set in order to provide an idea of the gap in performance resulting from removing parts of test images via occlusion. We highlight the performance reported in the last row of the Table. In that case, explanation maps for the group monotonic model are computed from only the features of the class slice, which is enough to match the performance of a standard ResNet-50 with full access to the features. This suggests that representations learned

| Model ($h$) | Aux. classifier | | | |
|---|---|---|---|---|
| | ResNext-50 | MobileNet-v3 | VGG-16 | SqueezeNet |
| Reference perf. | 77.62% | 74.04% | 71.59% | 58.09% |
| ResNet-50 | 72.94% | 68.31% | 67.34% | 49.95% |
| *Mono*ResNet-50 | 72.88% | 68.75% | 66.99% | 48.92% |
| *Mono*ResNet-50 (Constrained) | 72.44% | 66.55% | 66.92% | 45.83% |

Table 9: Top-1 accuracy of auxiliary classifiers evaluated on data created by occluding patches deemed irrelevant by explanation heat-maps given by different models. The performance of monotonic classifiers when constrained to consider only the feature maps within the slice corresponding to their prediction is further reported and shown to closely math the performance of cases where the full set of features is considered.

by group monotonic models are such that all the information required to explain a given class is contained in the slice reserved for that class.

## G EXAMPLES OF EXPLANATION HEAT-MAPS AND OCCLUDED DATA

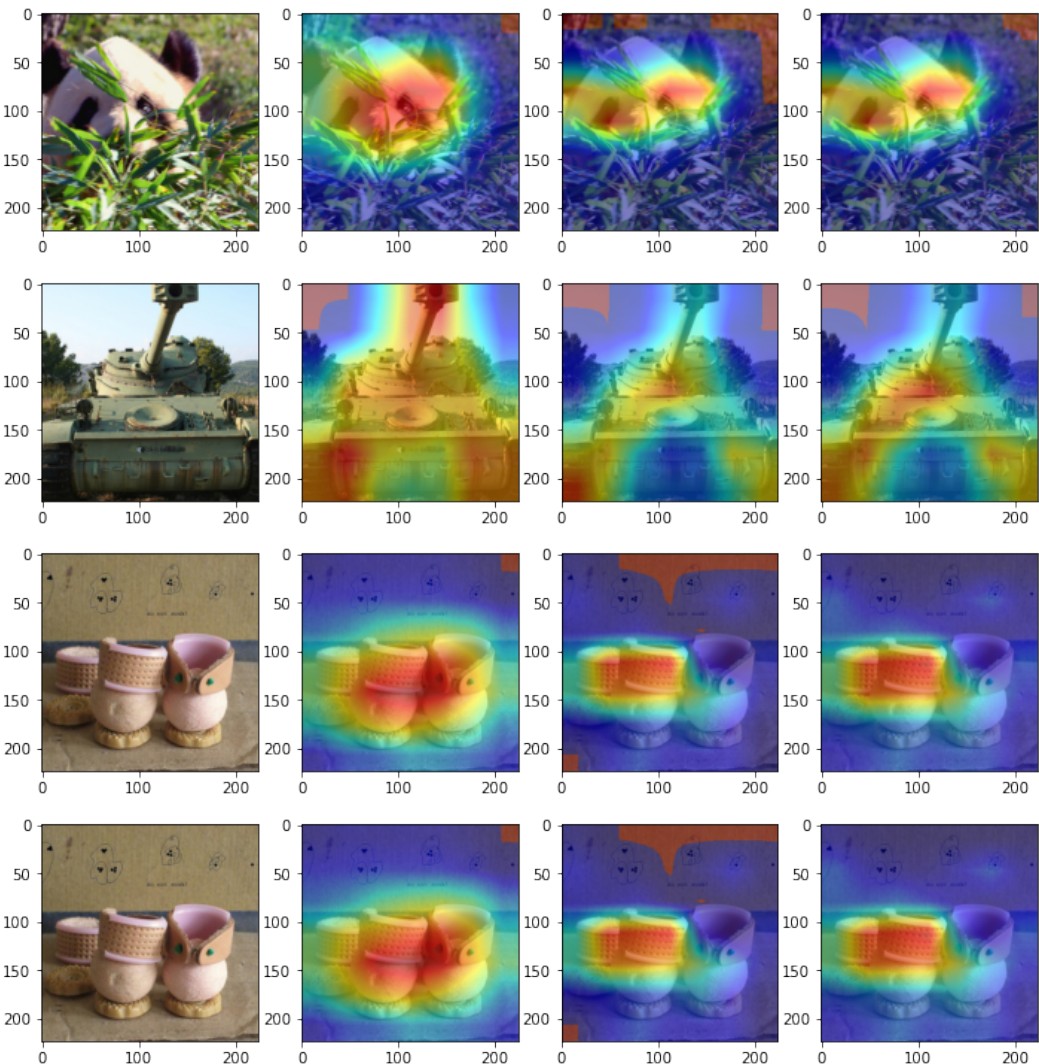

Figure 6: Examples of explanation heat-maps superimposed onto images. From left to right we have the original image, results obtained from a ResNet-50, a *mono*ResNet-50, and a *mono*ResNet-50 where the CAM operator only access the slice corresponding to the underlying class. All are obtained with Grad-CAM.

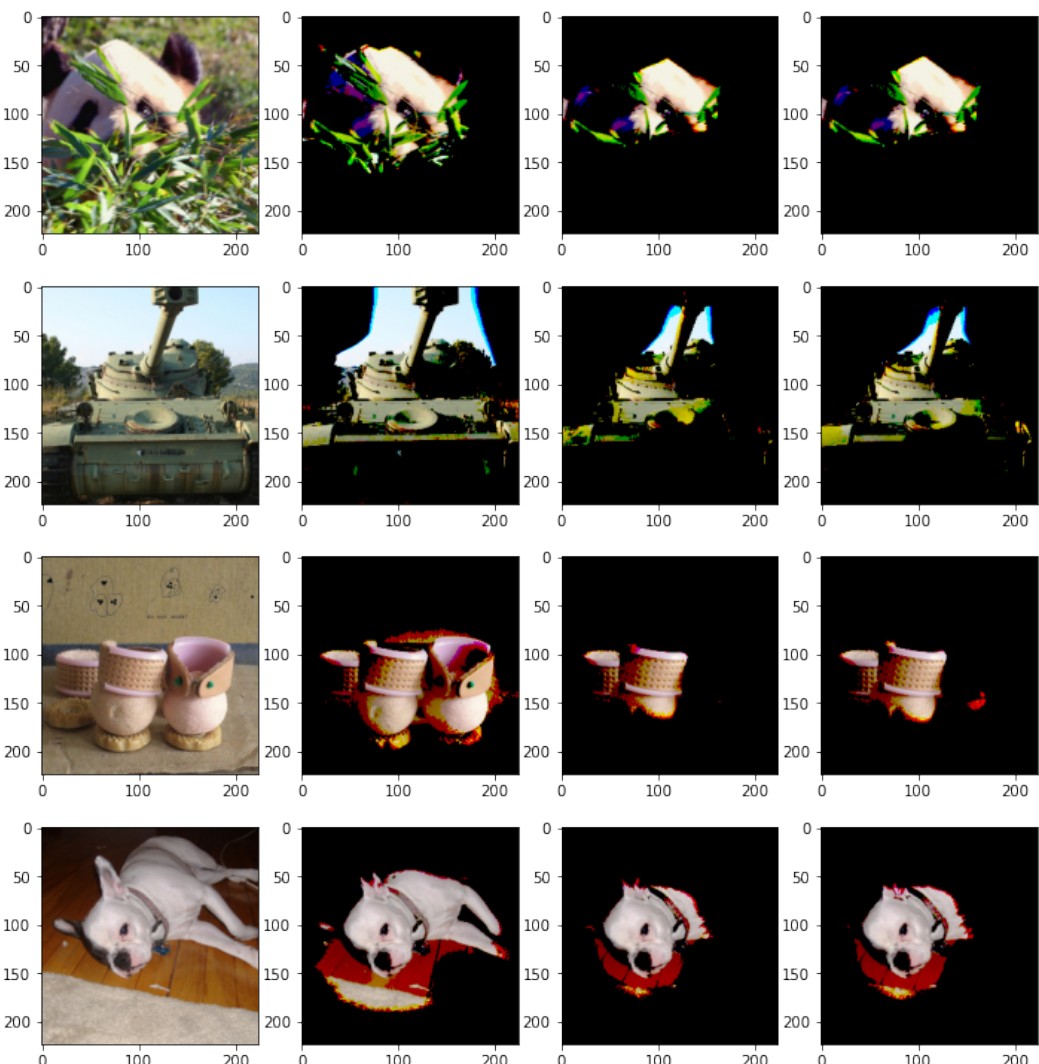

Figure 7: Examples of occluded data using explanation heat-maps. From left to right we have the original image, results obtained from a ResNet-50, a *mono*ResNet-50, and a *mono*ResNet-50 where the CAM operator only access the slice corresponding to the underlying class. All are obtained with Grad-CAM.

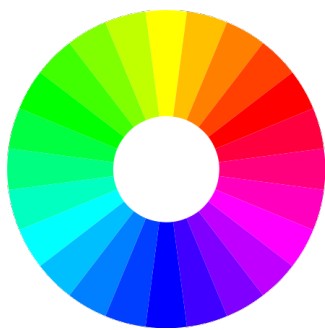

Figure 8: HUE circle of RGB images. Original image from: `https://en.wikipedia.org/wiki/Hue`.

## H ANALYSIS OF COLOR SEQUENCES FOR GENERATED DATA

We performed a set of experiments in order to evaluate whether some kind of ordering could be observed once we generate data for increasing values of $z$, specifically on dimensions that correspond to colors. To do that, we created an increasing sequence of values by defining a uniform grid in $[0, 1]$ with 50 steps. We then encoded a particular image, but decoded latent vectors after substituting the $z$ value in the dimension corresponding to *floor color* by the values in the sequence.

Generated sequences of images are shown in Figures 9 and 10 for the base and monotonic models, respectively. In each such a case, we plot the images on the left, and bottom-left patches of size 10x10 so as to highlight the color sequences that we observe with such an approach. Surprisingly, we observed that monotonic models tend to generate colors in a sequence that matches the HUE circle for RGB images, represented in Figure 8 for reference. Besides visually verifying that to be the case across a number of generated examples, in Table 4 in Section 4.3 we check the fraction of the dataset where such sequences of patches are sorted in terms of their HUE angles.

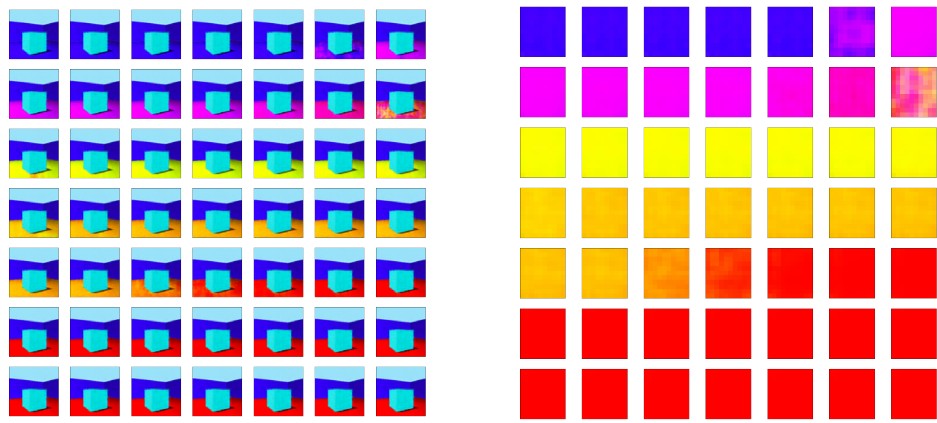

(a) Data for increasing values for the latent dimension associated to *floor color*.

(b) Bottom-left 10x10 patches of generated images.

Figure 9: Data generated by *standard model* for traversals of $z$ on the dimension corresponding to *floor color*

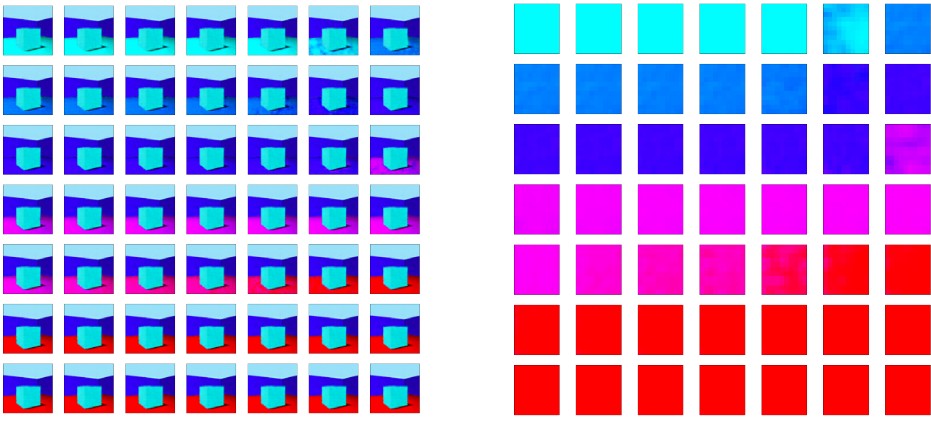

(a) Data for increasing values for the latent dimension associated to *floor color*.

(b) Bottom-left 10x10 patches of generated images.

Figure 10: Data generated by *monotonic model* for traversals of $z$ on the dimension corresponding to *floor color*

# I EXAMPLES OF DATA GENERATED WITH STANDARD AND MONOTONIC MODELS

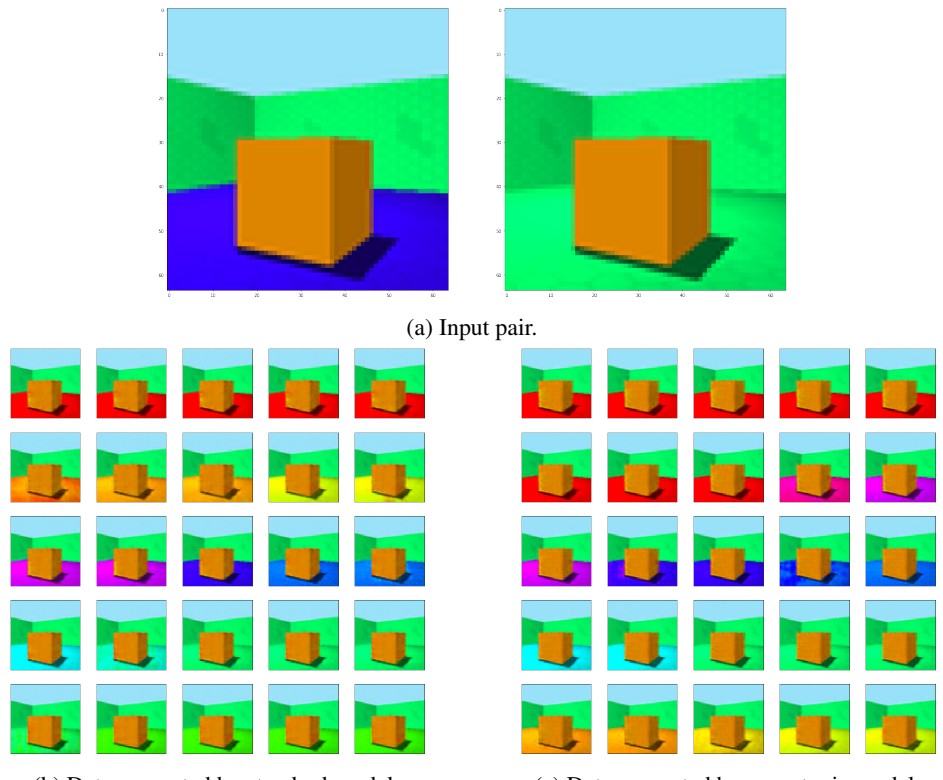

(a) Input pair.

(b) Data generated by standard model.  (c) Data generated by monotonic model.

Figure 11: Generating data by moving along the line passing over latent representation for inputs for which a single factor is different. Generative factor changing: floor color.

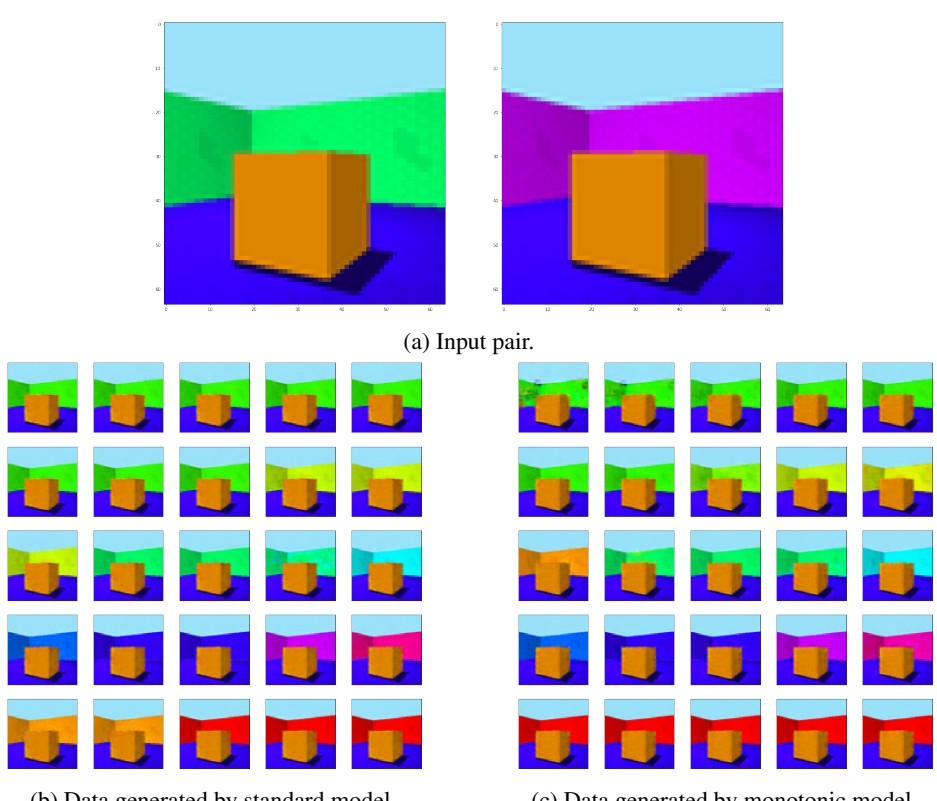

(a) Input pair.

(b) Data generated by standard model.      (c) Data generated by monotonic model.

Figure 12: Generating data by moving along the line passing over latent representation for inputs for which a single factor is different. Generative factor changing: wall color.

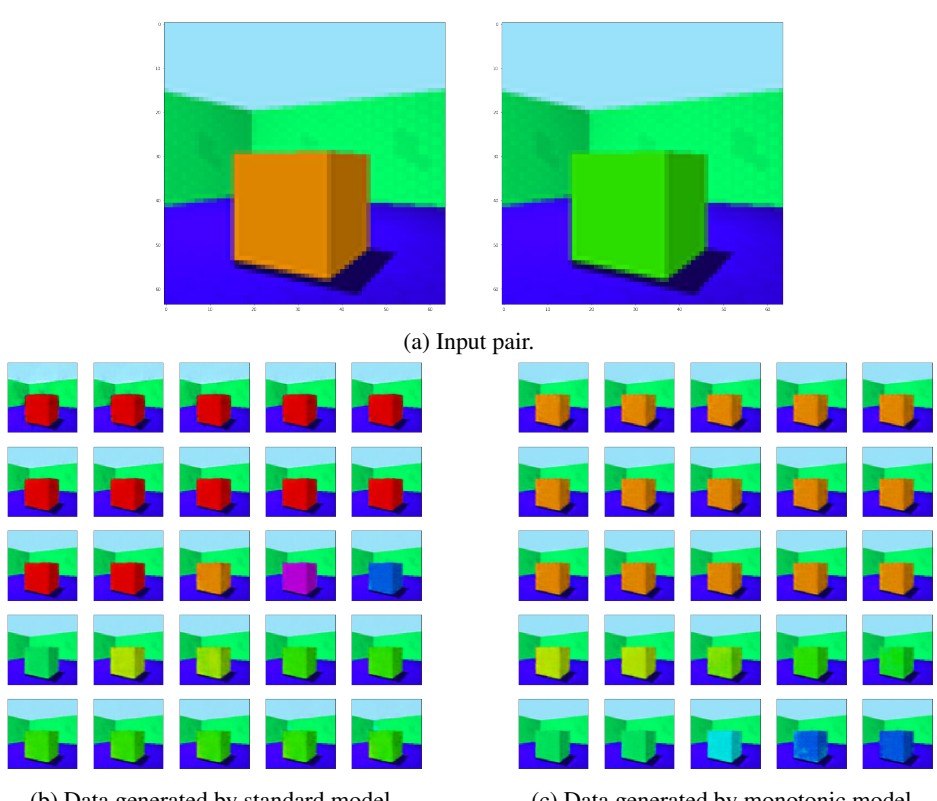

(a) Input pair.

(b) Data generated by standard model.  (c) Data generated by monotonic model.

Figure 13: Generating data by moving along the line passing over latent representation for inputs for which a single factor is different. Generative factor changing: object color.

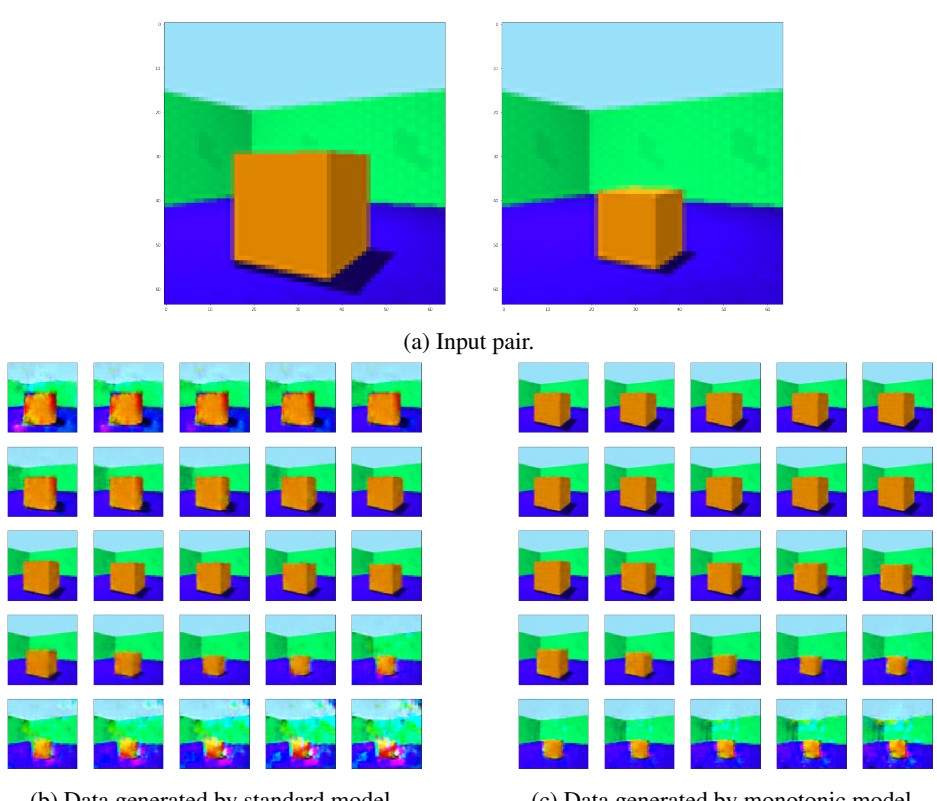

(a) Input pair.

(b) Data generated by standard model.    (c) Data generated by monotonic model.

Figure 14: Generating data by moving along the line passing over latent representation for inputs for which a single factor is different. Generative factor changing: scale.

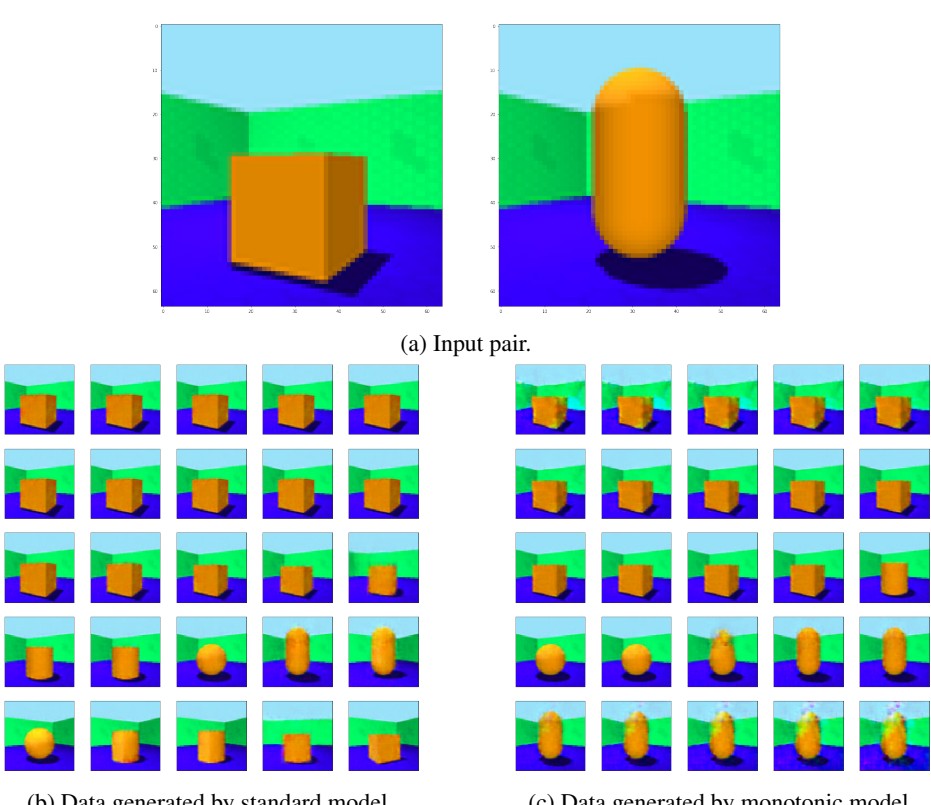

(a) Input pair.

(b) Data generated by standard model.

(c) Data generated by monotonic model.

Figure 15: Generating data by moving along the line passing over latent representation for inputs for which a single factor is different. Generative factor changing: shape.

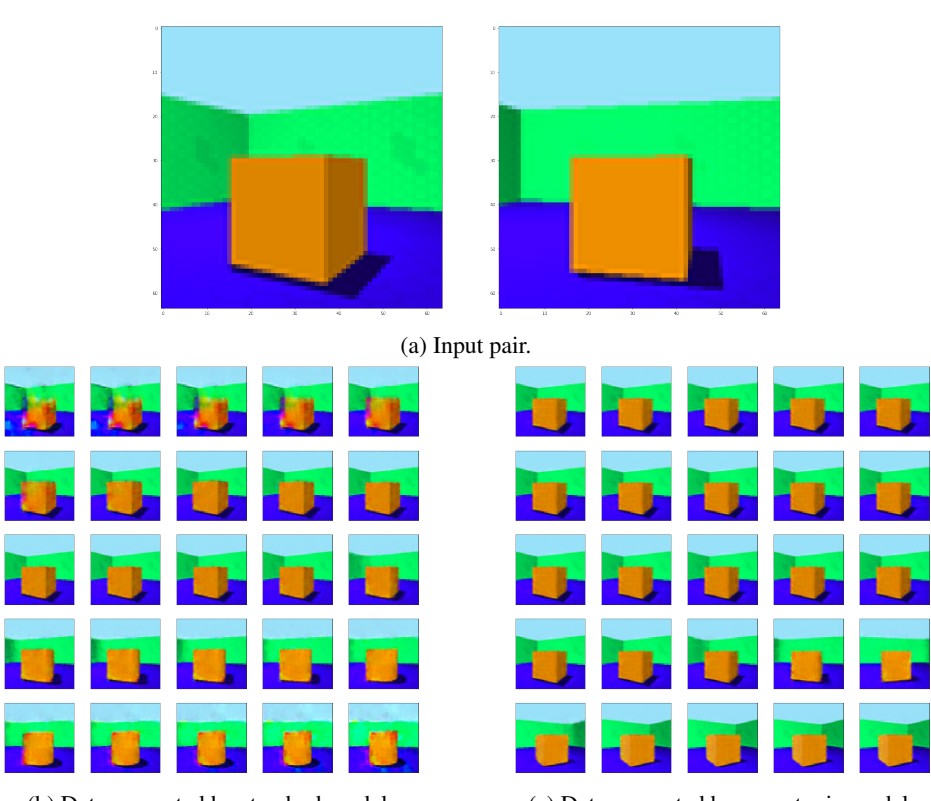

(a) Input pair.

(b) Data generated by standard model.          (c) Data generated by monotonic model.

Figure 16: Generating data by moving along the line passing over latent representation for inputs for which a single factor is different. Generative factor changing: orientation.

