# OpenReview forum: "Monotonicity as a requirement and as a regularizer: efficient methods and applications"
_ICLR.cc/2022/Conference — ICLR 2022 Submitted_

### Official Review · Reviewer_P7xG · 2021-10-28

**Correctness:** 3
**Technical Novelty And Significance:** 2
**Empirical Novelty And Significance:** 2
**Recommendation:** 5
**Confidence:** 4

**Main Review:**

$\textbf{Strengths}$

The paper is pretty straightforward and easy to understand. The content is supported with proper experimentation and detailed explanations. The authors demonstrate how to use monotonicity through penalty terms in models that are not constructed to be monotonic without the use of constraints.

$\textbf{Weaknesses}$

1. Consistent grammar issues.

Examples:

page 2 "such group of methods" --> "such a group of methods"

page 2 "monotonicity has been also observed" --> "monotonicity has also been observed"

page 2 "we define a new notion of monotonicity which is shown to be useful" --> "we define a new notion of monotonicity and show that it is useful"

page 2 "such property" --> "such a property"

page 2 "such class of models" --> "such a class of models"

page 8 "For example, assume we are interested in generating images of simple geometric forms, and desire to control factors..." --> "For example, assume we are interested in generating images of simple geometric forms and desire to control factors…"

There are also missing commas and other similar grammar issues.

2. Claim that lattice models "scale poorly with respect to the input space" is only partially true. This is the case for standard lattice models; however, the cited paper Monotonic Kronecker-Factored Lattice is an extension of lattice models that has linear space/time complexity. Thus, the claim in this paper that they scale poorly is not correct and needs to be changed. This claim sets the stage for demonstrating why the approach in the paper is desirable, but the claim is not true and the reality of linear space/time complexity hurts the argument of the paper. I would suggest re-working this claim to better show the reader why we would seek to use the methods outlined instead of a model that is monotonic by construction (and thus guaranteed to be monotonic everywhere in the input space and hold true even under a distribution shift).

3. Formatting issue section 3.1 the first sentence is weirdly formatted and is difficult to parse.

4. I found it a bit odd that the formula for mixup was not included in section 3.1 like the other penalty terms. I think explicitly writing out the formula for p_test like p_train and p_random would make things more clear for the reader.

5. Paper claims that performance is matched when imposing group monotonicity, but the results in Table 2 clearly show a significant decrease in performance. I would like the authors to more clearly explain their claim that performance is matched.



**Summary Of The Paper:**

This paper proposes techniques for using monotonicity both as a requirement and as a regularizer. The primary contributions are the mixup penalty term and group monotonicity, both of which build upon previous work in the field.

**Summary Of The Review:**

My primary concern with the paper is that Monotonic Kronecker-Factored Lattice provides a linear space/time complexity model that is monotonic by construction and thus monotonic everywhere in the input space and holds true even under a distribution shift. I believe that this model somewhat undermines the usefulness of the models described in the paper. I think the paper should make more clear why we would prefer to use the methods described over a model such as Monotonic Kronecker-Factored Lattice given that the model does scale well wrt the input space. This is not to say that the methods described are not useful; rather, this is to say that it should be made more clear why the methods described are as useful as claimed.

As for group monotonicity, I think this is a novel contribution that the authors have shown to work well and potentially be quite useful, particularly for the generative and adversarial cases.

For these reasons, I believe the paper is marginally below the acceptance threshold. Should the authors properly address my concerns, I see so reason not to bump up my score.

---

> ### Author Response · Authors · 2021-11-13
> **Response to reviewer.**
>
> Thank you for carefully reading our paper and for the useful feedback. We modified the manuscript accordingly. Also, please find our point-by-point responses below:
>
> **Q1. “Consistent grammar issues.”**
>
> A1. We addressed all grammar issues. Thank you for the detailed feedback.
>
> **Q2. “Claim that lattice models scale poorly with respect to the input space is only partially true.”**
>
> A2. We agree with the reviewer. We reworded the related paragraph accordingly. We now mention that lattice models generally scale poorly, but recent variations introduced efficient alternatives. However, we sustain that practitioners might still need to induce monotonicity in more general model classes, and that’s where our contribution fits in. In particular, approaches such as lattice models handle the monotonicity constraint effectively as it guarantees monotonicity by design. However, in case a particular application requires the use of some of the well-known architectures in deep learning, regularization-based strategies would be a viable alternative. Both of these two directions have pros and cons and the choice among them is application-dependent. In our contribution, we propose improvements to fix existing problems with the regularizer-based methods.
>
> **Q3. “Formatting issue section 3.1 the first sentence is weirdly formatted and is difficult to parse.”**
>
> A3. Fixed.
>
> **Q4. “I found it a bit odd that the formula for mixup was not included in section 3.1 like the other penalty terms.”**
>
> A4. We introduced a pseudocode in the Appendix C with a detailed explanation of our proposal from section 3.
>
> **Q5. “Paper claims that performance is matched when imposing group monotonicity, but the results in Table 2 clearly show a significant decrease in performance.”**
>
> A5. We clarify that, in that case, we mean to compare the prediction performance at the output layers given by standard vs. group monotonic classifiers. That is, we compare table entries in the rows for the left column. We thus claim that enforcing group monotonicity did not affect the prediction performances significantly. More specifically, for CIFAR-10, we got 95.46% vs. 95.64%, and for ImageNet we observed 75.85% vs. 76.50%.

---

> > ### Comment · Reviewer_P7xG · 2021-11-19
> > **Response to authors**
> >
> > Thank you for your response! You've addressed all of the concerns I had that were not covered by other reviewers. I would still like to note that, in practice, even a 0.2% drop in accuracy can be extremely impactful downstream. I think it would be worth mentioning explicitly why this decrease in performance is ok in relation to the added benefit of monotonic regularizers. Models that are constructed to be monotonic such as lattice models guarantee monotonicity, so a drop in accuracy is often worthwhile due to this guarantee and it's added benefits, but with regularizers we do not have such a strong guarantee. I would like some discussion around this.

---

> > > ### Author Response · Authors · 2021-11-20
> > > **Response to reviewer's comments.**
> > >
> > > Thank you for pointing that out!
> > >
> > > We reworded the discussion of results in table 2 and, rather than mentioning that the effects in accuracy are not significant, we now state that, although small, there were variations in the prediction performance when we included monotonicity penalties. However, we think that for the applications where monotonicity is a hard constraint, fine-tuning can achieve better performance as this is the right class of models to use.
> > >
> > > Moreover, we further clarify that including the penalty could also benefit performance. Specifically, results for ImageNet reported in table 2 show an improvement for the group monotonic model relative to standard classifiers. More specifically, for CIFAR-10, group monotonicity achieved a slightly lower accuracy of 95.46% vs. 95.64% for the baseline. On the other hand, for ImageNet, group monotonicity resulted in a higher accuracy of 76.50% vs. 75.85% for the baseline.
> > >
> > > Finally, similarly to lattice networks, regularizer-based models can guarantee monotonicity by applying a certification step as done in [1]. The certification step solves an optimization problem to find the minimum gradient. If the minimum gradient is positive, then monotonicity is guaranteed. If the trained network fails to pass the certification step, the monotonicity penalty is increased and the network will be retrained. This cycle is repeated until the network passes the certification step. The approach in [1] uses random data to apply the monotonicity penalty which might require multiple retraining steps. This emphasizes the importance of our approach in managing to find monotonic networks that pass the certification test in fewer iterations.”
> > >
> > > We hope this addresses the reviewer's remaining concerns.
> > >
> > > [1] Liu, X., Han, X., Zhang, N., & Liu, Q. (2020). Certified monotonic neural networks. arXiv preprint arXiv:2011.10219.

---

### Official Review · Reviewer_onzS · 2021-11-01

**Correctness:** 2
**Technical Novelty And Significance:** 3
**Empirical Novelty And Significance:** 1
**Recommendation:** 5
**Confidence:** 3

**Main Review:**

Strengths.
--------------
I found the paper written clearly and fairly easy to understand.
The claims in Sections 1--3 are justified and the authors provide good context for their work. The results of Section 3 are interesting for applications were partial monotonicity is desired but is not a hard requirement (e.g. for robustness and interpretability).
I found the application in Section 4.3 interesting. Showing how imposing monotonicity allows better controllability of a generative model.

Weaknesses
-----------------
1. In Section 3. The paper uses a measure $\rho$ that is essentially the fraction of examples at which *local* monotonicity (in any of the prescribed directions in $M$) is violated and then show that this measure decreases when using the paper's method over the baselines. However, I'm not certain that this measure corresponds to the global monotonicity requirment that is often desired in practice: namely, the one that appears in Definition 1. For example, consider a $1$-D function over $[1,99.99]$ whose graph is a piecewise linear curve connecting the points $(0,100), (0.99,100.99), (1,99), (1.99, 99.99), (2,98),  (2.99,98.99), ..., (99, 1), (99.99,1.99)$. This function has nonnegative derivative at about $99%$ of its domain, yet if one chooses two points $x_1, x_2$ uniformaly and independently from the domain, then there's at least a $97%$ chance that $f(min{x_1,x_2}) > f(max{x_1,x_2})$. I think, therefore, that it would be good to complement the local $\rho$ with an estimate of the probability that Definition 1 would not hold over the distribution in question (training, test or random).

2. Section 4.1
The authors introduce the notion of group monotonicity, but it's unclear how the regularizer introduced in equation 3 helps to encourage that property. Specifically, 1) Only the sum of the gradient is taken into account (so it could be that a component a_w_{i,j} has a very negative gradient, but still the sum will be positive), and 2) the softmax in equation 3 seems to encourage that the total gradient of $S_y$ is larger than the total gradient of all the other $S_k$'s, not that it's positive.
Perhaps I'm missing something?

3. Section 4.2
The paper claims that the fact that a good performance of the "total activation classifier" shows evidence that the original classifier satisfies group monotonicity.
But that claim is not clear to me. The total activation classifier does not depend on the part of the network that computes the output from the intermediate layer which is critical for the satisfaction of group monotonicity.

4. Section 4.2.2
The paper doesn't compare their methods to other methods for detecting noisey/adverserial test examples.



**Summary Of The Paper:**

The paper has two main contributions:

1. It takes a known monotonicity regularizer and trains with it using a new distribution that is roughly a mixing of uniform and the training distribution. It shows empirically that using this method increases the "size" of the input region in which the model is monotonic.

2. It defines several regularizers that are meant to encourage "monotonic behavior" of the model's output w.r.t the outputs of an intermediate layer or latent variable. It provides experiments that show how using these regularizers does not hurt the model's performance and two applications of the added structure: 1) detecting noisey/adverserial examples and 2) more controllable generation in generative models.


**Summary Of The Review:**

See above.

---

> ### Author Response · Authors · 2021-11-13
> **Response to reviewer.**
>
> We would like to thank the reviewer for a careful reading of our paper and for providing constructive feedback. Please find our point-by-point responses below:
>
> **Q1. “In Section 3. The paper uses a measure  that is essentially the fraction of examples at which local monotonicity...I think, therefore, that it would be good to complement the local \rho with an estimate of the probability that Definition 1 would not hold over the distribution in question (training, test or random).”**
>
> A1. If the gradient is greater than 0 for all x in the input domain, then this guarantees that the function is monotonically non-decreasing. As for using a finite set of points to test the monotonicity condition, the approach in [1] uses random data to apply the monotonicity penalty. After training, a certification step that checks for global monotonicity is required. This is computationally costly, especially for large networks. So, as a work around, previous works have proposed to either enforce monotonicity on the training data or on random points. Our work shows that you get monotonicity where you enforce it. As such, the work using the training data fails on random points and the work using the random points fails on the data. Our method covers both. However, there could still be counter examples that we can find violating monotonicity. The chances are just lower using our method because we are covering a bigger part of the space.
> The counter-example provided by the reviewer is very interesting. However, we think it actually supports our line of work. We need methods that check for the gradient in more important parts of the space given a fixed computational budget. Our method is more likely to capture the negative gradient in the intervals (0.99,1), (1.99,2),...,(89.99,99) compared to methods relying only on random points or the training data. In practice, our proposal can thus reduce the number of certification and retraining steps required to guarantee monotonicity.
>
> **Q2. “...Only the sum of the gradient is taken into account...the softmax in equation 3 seems to encourage that the total gradient...”**
>
> A2. The reviewer is completely correct in both assessments: there could be negative gradients within a class slice even if group monotonicity is satisfied, and the cross-entropy nature of the regularizer enforces that the total gradient of a specific slice is greater than the others, but they could all be negative. However, we remark that group monotonic classifiers need not verify monotonicity per requirement, and we have more flexibility in this case. We thus take advantage of this flexibility to define regularization strategies that are easier to implement and train against. In particular, under similar concerns as those raised by the reviewer, we highlight that during our experiments we experimented with penalties that focused only on the negative components of the gradients. Doing so, however, made it much harder for the penalty to be minimized, required larger regularization weight, and affected prediction performance. All of that was fixed when we moved to the softer versions presented in the paper. Moreover, in all of our experiments, towards the end of training, all of the units within the slice corresponding to a given class become positive. Our decision was then motivated by those empirical observations.
>
> **Q3. “The total activation classifier does not depend on the part of the network that computes the output from the intermediate layer which is critical for the satisfaction of group monotonicity.”**
>
> A3. We clarify that the total activation classifiers are defined using the same slices of the features as those used to define group monotonicity penalties. In doing so, when we enforce group monotonicity, we structure the features so that each slice will “fire up” more strongly than the others depending on the underlying class.
>
> **Q4. “The paper doesn't compare their methods to other methods for detecting noisey/adverserial test examples.”**
>
> A4. We remark that we do not seek to introduce adversarial examples detectors in this work. The experiment reported in Table 3 is intended to provide evidence supporting the claim that simple statistics of features provided by group monotonic models are effective in indicating anomalous inputs. Training state-of-the-art level detectors would require some sort of adversarial training as well as other design variations that we believe would go beyond the scope of this work. Nonetheless, the experiment in question provides clear insights in terms of how effective verifying certain imposed structures in the features can be for detecting anomalies, which opens up new research directions.
>
> [1] Liu, X., Han, X., Zhang, N., & Liu, Q. (2020). Certified monotonic neural networks. arXiv preprint arXiv:2011.10219.

---

> > ### Comment · Reviewer_onzS · 2021-11-19
> > **Response to authors**
> >
> > Thank you for the clarifications!
> >
> > Here are my specific replies:
> > Q1. Thank you for pointing out that the "fraction of places where the gradient is positive" metric is based on prior work. However, I still think (and as my example shows) that this metric is not a good proxy for the practical definition of monotonicity. To really evaluate your methodology, I'd like to see an empirical estimate of the probability that Definition 1 would not hold over the distribution in question (training, test or random). This obviously does not need to be done in practice when using the method (since it is expensive as you remark), but it is important for evaluating it in a research paper.
> >
> > Q2. Thank you for explaining how you arrived at the current penalty. Since the penalty seems to only encourage monotonicity of some of the gradients, it doesn't look like group monotonicity is the right definition of the property that makes the classifier perform well, here. I think further thought is required to define the phenomena better, here.
> >
> > Q4. Thank you for clarifying. This resolves my previous remark here.

---

> > > ### Author Response · Authors · 2021-11-20
> > > **Further clarifications on the reviewer's comments.**
> > >
> > > Thank you for going through our responses and providing further feedback! We would like to provide extra clarifications in order to address the raised concerns:
> > >
> > > **Q1. "a good proxy for the practical definition of monotonicity."**
> > >
> > > A1. While we thank the reviewer for the suggestion, we discuss this point further and raise some practical concerns in the proposed evaluation metric:
> > >
> > > - Definition 1 and a local monotonicity definition based on the gradients are equivalent. We can choose one or the other without any compromises. We have decided to use the definition using the gradient following [1]. We checked the gradients using a finite set of points and if we have good coverage of the space, then we can better detect counter-examples.
> > > Definition 1 was more explicitly used in [2] to evaluate the performance. However, they restrict the type to pairs of points close to the training data in order to make the algorithm more efficient.
> > >
> > > - Computing a metric based on Definition 1 would require access to pairs of points where input dimensions only change across monotonic dimensions, while non-monotonic dimensions are kept fixed. We could define sampling strategies where we sample a point and then change only the monotonic dimensions, but that would hardly be efficient. Problems are, for example: how to change the monotonic features? Should we sample them randomly? Or do some perturbations? In all such choices, the cost to evaluate the metric would scale exponentially with the number of monotonic dimensions. Also, there could still be counter-examples and we would not have more evidence of monotonicity than what we already have via \rho.
> > >
> > > - We propose an example where the gradient approach is more efficient in finding counterexamples than using a pair of points. For example, consider a 1-D function over [0,100] whose graph is a piecewise linear curve connecting the points (0,0),(0.99,1.99),(1,1),(1.99,2.99),(2,2),(2.99,3.99),...,(99.99,100.99),(100,100). This function has a 1 percent chance that the gradient is negative. However, to detect that the function is not monotonic, we need 2 points to lie on the small segment that has negative gradient. The chance here is 0.01*0.01= 0.01 percent. In this case, the gradient approach is more efficient.
> > >
> > > - To fully guarantee monotonicity, we need to apply a certification step as done in [1]. The certification step solves an optimization problem to find the minimum gradient. If the minimum gradient is positive, then monotonicity is guaranteed. If the trained network fails to pass the certification step, the monotonicity penalty is increased and the network will be retained till the network passes the certification step. The approach in [1] uses random data to apply the monotonicity penalty which might require multiple retraining of the network. This emphasizes the importance of our approach as having smaller \rho would mean that we can find monotonic networks that pass the certification test in fewer iterations.
> > >
> > > **Q2. "Since the penalty seems to only encourage monotonicity of some of the gradients, it doesn't look like group monotonicity is the right definition of the property that makes the classifier perform well, here."**
> > >
> > > A2.
> > >
> > > - We would like to clarify that group monotonicity is not a hard constraint that needs to be satisfied. This property was inspired by monotonicity and the goal here is to impose structure in the features rather than to satisfy a constraint imposed by the downstream application. So we have more flexibility in enforcing the constraint.
> > >
> > > - We used this flexibility in order to develop an efficient training procedure. The proposed penalty does encourage group monotonicity. The goal was to have a distinct slice of feature maps activated for each class.
> > >
> > > - While the penalty could be minimized with non-group monotonic classifiers (all gradients negative, but the right slice fires up more strongly than the others), those do not tend to happen. In fact, using the reported penalty, 100% of our training runs across all datasets resulted in models for which the minimum gradient in the feature slice corresponding to the given label is positive (We are happy to add figures with those statistics to the appendix if the reviewer feels they will add value).
> > >
> > > - Table 2 provides clear evidence that total activation classifiers induced by group monotonic models perform closely to the performance given by the classifier defined at the output layer, which could only be achieved via group monotonicity.
> > >
> > > We hope this helps clarify the remaining concerns of the reviewer and we will be happy to provide further details or further edit the manuscript to make it clearer.
> > >
> > > [1] Liu, X., Han, X., Zhang, N., & Liu, Q. (2020). Certified monotonic neural networks. arXiv preprint arXiv:2011.10219.
> > > [2] Sivaraman A, Farnadi G, Millstein T, Van den Broeck G. Counterexample-Guided Learning of Monotonic Neural Networks. Advances in Neural Information Processing Systems. 2020;33:11936-48.

---

> > > ### Comment · Reviewer_onzS · 2021-11-24
> > > **Response to Authors (cont.)**
> > >
> > > Thank you for the clearly thoughtful response.
> > > However, my remaining concerns with the paper still hold. Specifically:
> > >
> > > Q1. I agree that (for differentiable functions) having the gradient be positive in *all* the domain is equivalent to Definition 1 holding in *all* of the domain. However, the example I provided as well as yours show that weakening one part of this equivalency slightly does not generally map to the same slight weakening of the other part. I.e., if the gradient is positive in "90%" of the domain it doesn't generally mean that Definition 1 would hold in "90%" of the main. Now, in my opinion, in applications where monotonicity is a hard requirement, having Definition 1 hold in most of the domain is more important than having the gradient be positive in most of the domain: e.g., taking your example of a model used to accept/reject job applications: I would think that in practice, a user of the model would care more that there is a low probability of seeing two applicants that differ only in the monotonic directions and have models outcome that contradict the monotonicity requirement, rather than knowing that most of the time small perturbations of a monotonic feature would not cause the outcome to change in a direction that contradicts the requirement. However, even if that's not true, both of our examples show, that you at least need both metrics to faithfully tell the story of "how much" a model is enforcing the monotonicity requirement.
> > >
> > > Q2. To clarify, my concern is in the exposition used to justify the "group monotonicity" property, not with the empirical results which I find essentially valid. The connection between the definition of "group monotonicity" and the regularizer used to encourage it is unclear and  seems a little contrived. It almost seems as if the regularizer is encouraging a related, yet different, property: namely, that the total gradient of $h(x)$ with respect to the slice $S_y$, where $y$ is the true label associated with input $x$, is the largest among the total gradients of $h(x)$ w.r.t other slices. I think that changing the exposition in Section 4 to report the unsuccessful attempts to use a regularizer that actually focuses on positivity of the gradient would benefit readers, and would help the reader to better understand the motivation.

---

> > > > ### Author Response · Authors · 2021-11-25
> > > > **Further clarifications on the reviewer's comments.**
> > > >
> > > > We thank the reviewer for the constructive and helpful feedback. Please refer to our comments below:
> > > >
> > > > **Q1. Evaluation metric for section 3:**
> > > >
> > > > A1. The reviewer suggested using/adding another metric that is based on the violation of monotonicity on pairs of points.
> > > > However, to the best of our knowledge, no such metric exists in the literature. Moreover, building pairs manually would scale exponentially with the number of monotonic dimensions. I.e., the choice of the initial points and how to do perturbations of the monotonic features to test Definition 1 can have many possibilities.
> > > >
> > > > That being said, if the reviewer would trust a specific recipe to build pairs of points with the desired property, we will be more than happy to evaluate it on at least one of the datasets we considered, and report results here and/or in the paper.
> > > >
> > > > **Q2. group monotonicity property**
> > > >
> > > > A2. Thank you for the clarification. We agree with the reviewer. We reworded the paragraph that introduces the penalty and mentioned that non-group monotonic models could minimize the proposed penalty, but we didn't observe those cases to happen in practice. We also included feature statistics in the appendix showing that using the current proposal, group monotonicity is achieved approximately halfway through training. We also mentioned that we evaluated cases that focused on negative gradients, but that did not work as well as the proposal. These changes will appear in the camera-ready version of the paper if it is accepted.
> > > >
> > > > We thank the reviewer once more for their contribution to our paper and hope to have addressed their concerns.

---

> > > > > ### Comment · Reviewer_onzS · 2021-11-29
> > > > > **Response to authors**
> > > > >
> > > > > Thank you for making the changes. However, I think the paper needs further work to demonstrate the effectiveness of their method. Thus, my initial rating remains unchanged.

---

> > > > > > ### Author Response · Authors · 2021-11-29
> > > > > > **Response to reviewer**
> > > > > >
> > > > > > Thank you for engaging with us in the discussion and for the suggestions; we believe they helped improve our paper significantly. We would greatly appreciate it if the reviewer would provide more specific feedback on exactly what is missing in our work. This way, we can improve the paper accordingly.

---

### Official Review · Reviewer_KKi2 · 2021-11-05

**Correctness:** 3
**Technical Novelty And Significance:** 2
**Empirical Novelty And Significance:** 2
**Recommendation:** 5
**Confidence:** 4

**Main Review:**

Strengths
1) This paper addresses an important problem.  The monotonicity constraint is often overlooked by the ML community, but it does play an important role in many real-world applications. Therefore, an improved method to encourage monotonicity can have sizable impact on these applications, and help ML practitioners better impose their priors on the hypothesis space.
2) Writing is clear.

Weaknesses
1) The contribution is rather incremental.
2) The authors didn't provide adequate explanations for certain important details in the paper itself.  For example, the authors propose to create synthetic examples by interpolating both features and vectors, but in the experiment involving binary classification data set, it's not clear how to interpolate the labels here. Did the authors simply interpolate between 0 and 1?  Why we need to use both interpolated examples as well as random examples (not clear from the paper)?

**Summary Of The Paper:**

This paper proposes an incremental improvement to existing methods that encourage monotonicity through a regularization term. The contribution of the paper is about how to sample the data to compute this regularization term, which is an expectation w.r.t. a data distribution. So instead of purely sampling from existing training data or performing uniform sampling in potentially high-dimensional feature space, this paper proposes to use mixup, which essentially involves creating synthetic examples by interpolating between existing training examples. The authors also extend this regularization term to the case where the prediction function outputs a vector, e.g., multi-class classification), and the case of VAE. The authors provide some experiment evidence that the proposed change can reach better performance for some data sets.

**Summary Of The Review:**

Given the incremental contributions, I consider this paper marginally below the acceptance threshold. It can be made stronger if the authors can provide more insights or make the experiments more illustrative or extend to the case of label's order on categorical features (treated as a partially ordered set).

---

> ### Author Response · Authors · 2021-11-13
> **Response to reviewer.**
>
> Thank you for your comments and valuable feedback. Please see below our detailed responses to the questions:
>
> **Q1. “The contribution is rather incremental.”**
>
> A1. We highlight that we do not use standard mixup as a regularization strategy. Rather, in the first part of the paper, we use mixtures of training data and random points to compute regularization penalties in regions that were ignored in past approaches.
>
> Overall, our contributions can be summarized as follows:
>
> 1) Identifying and explaining issues in recent literature: We show that sampling schemes used to implement monotonicity regularizers actually affect the behavior of resulting models, and should be done carefully.
>
>
> 2) Proposing simple, easy-to-use, and efficient approaches to address those issues: We propose a novel application of mixup (i.e., mixing data with noise) to induce non-uniform distributions that we can sample from efficiently and that populate parts of the space that would be overlooked by previous approaches. We provide comprehensive empirical evidence supporting the claim that the proposal fixes issues in previous methods.
>
>
> 3) Extending the notion of monotonic models to larger-scale cases: We introduce regularization strategies for image classifiers and generative models that enforce a certain structure in learned features, which can be leveraged for several applications. We specifically highlight the case of controllable generative modeling. The baseline plain model from Locatello et al. (2020) [1] achieves disentangled representations, but it is hard to decide how to change the latent code to control the data generation. On the other hand, our model is disentangled while enabling more controllable generation. For example, in our model, as we increase the value of the corresponding latent factor, the colors generated follow the HUE order, the order of the generated shapes is consistent, and the size of the generated objects goes from small to big. To the best of our knowledge, this is the first work that identified this limitation in factorized VAEs and proposed efficient solutions.
>
> We included Algorithm 1 in Appendix C further detailing the procedure we use to compute mixup-based regularization penalties.
>
> **Q2. “it's not clear how to interpolate the labels here. Did the authors simply interpolate between 0 and 1?”**
>
> A2. This part highlights another contribution of our work. We clarify that we do not use the standard mixup idea between data points as it was originally intended for. Rather, in our case, mixup is used in a regularization penalty to enforce monotonicity. Our solution performs mixup between data and random points to better select points on which the monotonicity penalty is evaluated. Thus, it is only used for computation of \Omega, where the labels are not required.
> We further note that data examples and random points are mixed together just so that the gradient penalties \Omega can be applied in parts of the space that would be neglected otherwise.
>
> **Q3. “Why we need to use both interpolated examples as well as random examples (not clear from the paper)?”**
>
> A3. We need to mix random points and data points; in doing so, we populate the space between the data manifold (approximated by the training points) and the boundaries of the input space (approximated by the random points). This is precisely the region that is ignored prior work, and mixup is an efficient approach to sample from distributions with support in those regions. We kindly refer the reviewer to appendix A for a more in-depth discussion of the effect of mixup when mixing data and boundary (random) points in high dimension. We included Algorithm 1 in Appendix C for more details of the procedure used to compute the mixup-based regularization penalties.

---

### Official Review · Reviewer_Sj4Q · 2021-11-06

**Correctness:** 3
**Technical Novelty And Significance:** 3
**Empirical Novelty And Significance:** 3
**Recommendation:** 6
**Confidence:** 3

**Main Review:**

Clarity:  Overall, the work was clearly written with a few minor typos here and there.  I felt that the group monotonicity section was a bit tough to follow at the beginning -- I only really understood what you were trying to do after seeing the applications.  It could probably be better motivated.

Novelty:  The hard constraint on monotonicity is a simple application of Mixup.  While the evaluation is definitely of general interest, the novelty is a bit low here.  The second approach is novel to me at least.  On first reading, I wasn't even certain what kinds of things that you could do with it.  I think it should really be explained a bit more in the introduction as it was, to me at least, a very different kind of requirement.

Significance:  The significance of this work, in my opinion is entirely in the new group monotonicity property.  While the proposed applications of this were interesting, I feel that there could possibly even more utility in certain domains -- though I can't say for certain.

Specific comments:

- "i.e." and "e.g." must always be followed by a comma, e.g., like this.
- As mentioned about, I'd really like to hear your high level motivation for the group monotonicity property.  It appears to be a bit different from the first type of monotonicity discussed, i.e., it's different than just soft enforcing a hard monotonicity constraint, right?
- What was the configuration of the detector in Table 3?  I didn't see an explanation of the choice of tau?  How did you determine the perturbation budgets?  Thoughts about why performance on MNIST is so good?
- Figure 2 was a bit small and too difficult for me to really get a good understanding of what it was trying to show.  I understand that some types of monotonicity result from the training, but it's less clear to me what advantages it has in this case (other than showing off the approach).

**Summary Of The Paper:**

The paper investigates two different strategies for adding monotonicity into the training of NNs: one in which the monotonicity is required on some subset of the input and one in which monotonicity is encouraged, but not required, in a part of the model.  Experiments illustrate that these approaches do achieve different types of monotonicity in practice.

**Summary Of The Review:**

An interesting paper that fills some gaps in existing work and posits some interesting applications of monotonicity.  The first contribution seems a little weak as it is built almost exclusively from existing methods with little deviation while the second contribution has more novelty.  I feel that the second approach needs some better motivation and possibly a better application  -- I would prefer that you drop the first approach to include it.

---

> ### Author Response · Authors · 2021-11-13
> **Response to reviewer.**
>
> Thank you for your encouraging comments and your positive feedback! In what follows, we would like to address raised concerns more carefully, point by point:
>
> **Q1. “The hard constraint on monotonicity is a simple application of Mixup.”**
>
> A1. We would like to point out that our contribution in section 3 goes beyond the introduction of mixup to compute monotonicity penalties. In particular, we identify major issues and shortcomings in the past approaches that were still unknown. For example, one of the main issues is that using either the training distribution or arbitrary easy-to-sample-from priors does not ensure the property will be enforced where it must be. Besides raising the issue and explaining how it manifests, we propose mixup as an efficient solution to address the issue. Rather than applying mixup on the data points, our solution proposes using mixup between data and random points to select points on which the monotonicity penalty is evaluated. We kindly refer the reviewer to the more in-depth discussion reported in appendix A, where we show how mixup operates and helps to fill the gap in the recent state-of-the-art methods.
>
> **Q2. “"i.e." and "e.g." must always be followed by a comma”**
>
> A2. Thank you for pointing this out. We fixed those typos throughout the text.
>
> **Q3. “I'd really like to hear your high level motivation for the group monotonicity property.”**
>
> A3. That is correct: group monotonicity is not a requirement. However, we argue that imposing the property is beneficial in different ways, and can be done efficiently, without affecting the wall-clock time to accuracy. The property introduces structure in the representations, and the underlying structure can be leveraged at testing time (e.g., defer prediction to human if the property is not observed during testing, select the right feature maps to compute visual explanations).
>
> We demonstrate the capabilities of group monotonicity in our experiments. More precisely, section 4.3 shows how we can use monotonicity in controllable data generation. In this case, the baseline plain model from Locatello et al. (2020) [1] achieves disentangled representations but it is hard to decide how to change the latent code to control the data generation. On the other hand, our model still learns disentangled representations while also allowing for controllable generation. For example, in our model, as we increase the value of the corresponding latent factor, the colors generated follow the HUE order, the order of the generated shapes is consistent, and the size of the generated objects goes from small to big.
>
> **Q4. “What was the configuration of the detector in Table 3?”**
>
> A4. Detection in Table 3 is implemented via the entropy of the total activation classifier induced by group monotonic models. In other words, we use the total activations of class slices as logits to define conditional categorical distributions over the label set. High entropy in such a distribution is indicative of anomalous inputs since the group monotonicity property is not verified (that is, there’s not only one feature slice “firing up” more strongly than the other slices for this input).
>
> **Q5. “I didn't see an explanation of the choice of tau?”**
>
> A5. We performed evaluations in terms of the area under the operating curve (ROC-AUC), which does not require the definition of a threshold. If we were to make actual predictions, then we would need to pick a specific \tau depending on the application specifications of precision and recall. We used the AUC implementation in sklearn: https://scikit-learn.org/stable/modules/generated/sklearn.metrics.auc.html
>
> **Q6. “How did you determine the perturbation budgets?”**
>
> A6. We used standard perturbation budgets from the adversarial attacks/defenses literature (e.g., [2]). Specifically, we used 0.3 and 8/255 for MNIST and CIFAR-10, respectively, both in terms of L_{\inf} distances.
>
> **Q7. “Thoughts about why performance on MNIST is so good?”**
>
> A7. There are two main reasons for that: 1) we observed that enforcing group monotonicity was much easier in smaller scale models/data compared to more complex architectures and datasets. That is to say that models trained on MNIST were “more group monotonic” (or induced less entropic/more confident total activation classifiers) than CIFAR-10 classifiers. 2) MNIST attacks are easier to detect (visually) than CIFAR-10 attacks (less complex images render artifacts in a way which is easier to spot).

---

> > ### Author Response · Authors · 2021-11-13
> > **Continuation of response to reviewer.**
> >
> > **Q8. “Figure 2 was a bit small and too difficult for me to really get a good understanding of what it was trying to show.”**
> >
> > A8. Factorized VAEs are often motivated as a means to allow for controllable data generation. We argue that factorization of the latents is necessary but not sufficient to enable controllability and that some notion of monotonicity is also required. In figure 2, we show the difference between data generated via traversals in the latent space. Models satisfying monotonicity present more regular (e.g., without cycles) and smoother transitions, and are hence more controllable than the standard factorized models. Specifically for factors related to colors, in appendix H, we show that monotonic models generate color sequences that, surprisingly, follow the HUE cycle. We will include larger versions of individual component of Fig. 2 in the Appendix.
> >
> > **Q9. I feel that the second approach needs some better motivation and possibly a better application -- I would prefer that you drop the first approach to include it.**
> >
> > Thank you for the suggestion. However, the first part of the paper includes results which we think are of relevance to the community. Specifically, we raise issues present in state-of-the-art monotonicity enforcing regularizers given by the fact that they limit the parts of the space where the property is actually enforced. We further introduce a simple fix to that issue in the first part of the paper.
> >
> > [1] Locatello, F., Poole, B., Rätsch, G., Schölkopf, B., Bachem, O., & Tschannen, M. (2020, November). Weakly-supervised disentanglement without compromises. In International Conference on Machine Learning (pp. 6348-6359). PMLR.
> >
> > [2] Zhang, H., Yu, Y., Jiao, J., Xing, E., El Ghaoui, L., & Jordan, M. (2019, May). Theoretically principled trade-off between robustness and accuracy. In International Conference on Machine Learning (pp. 7472-7482). PMLR.

---

### Author Response · Authors · 2021-11-22
**Summary of updates and discussion with reviewers.**

We thank all the reviewers for their comments! We believe the provided feedback helped us improve the manuscript and made the presentation clearer. In what follows, we highlight the changes in the paper, the main points of discussion, and summarize once more our contributions:

**Changes:**

- Included algorithm 1 in the appendix in order to clearly define the mixup approach we introduced.

- Modified discussion of related work to point out that there are efficient variations of lattice models, but regularization-based approaches are required for applications where general model classes need to be used. In this case, certification approaches are required in order to fully guarantee monotonicity.

- Modified the discussion of group monotonic results to point out that the penalty has a small effect on prediction accuracy. We highlight that in some cases it improved performance (ImageNet) and in another case (CIFAR-10) it resulted in a slight loss of accuracy.

- Modified section 3.1 to make it more readable.

- We fixed typos throughout the text and in the definitions of group monotonic penalties.

**Summary of discussion:**

- Reviewer onzS:

    - The reviewer questioned the validity of \rho as a good validation metric and suggested an alternative. We pointed out the reviewer's proposal and \rho are equivalent (for differentiable functions), and both options are prone to counter-examples. The chosen \rho, however, can be implemented much more efficiently since evaluation based on pairs requires a number of points that scale exponentially with the number of monotonic dimensions.
        - We further remark that \rho *measures the probability of a model to fail in a monotonicity certification*. Our regularization strategies consistently reduce such a metric, indicating that it effectively results in models more likely to be certifiably monotonic.
        - Results in table 1 correspond to confidence intervals obtained from 20 independent training runs of each approach in each dataset. Moreover, \rho is evaluated with a rather large sample (all the available training/testing data, or samples of size 10000 for the random case). We believe this renders the evidence in table 1 robust and indicates clear benefits of mixup relative to past proposals (e.g., compare to the control case where no monotonicity penalty is applied).

    - The reviewer highlighted the fact that non-group monotonic predictors could minimize the proposed penalty from section 4. We highlighted that those cases simply do not happen in practice and if they did, it wouldn't be an issue since those are employed for applications that don't have monotonicity as a hard constraint. Besides the results in Table 2, we are happy to include an appendix with more evidence for the fact that group monotonicity is achieved with the proposed penalty if the reviewers deem it relevant.

- Reviewer P7xG:

    - The reviewer pointed out that there are efficient lattice models. We fixed our discussion of prior work.

    - The reviewer pointed out that the variations in the accuracy we observed could be relevant in certain applications. We fixed our discussion of results to indicate that. We remark though that we observed mixed effects and the proposed penalty sometimes slightly improves performance while the opposite happens in some cases.

**Summary of contributions:**

- We present problems that were overlooked in past literature on training monotonic models using regularization schemes.

- We provide a practical, inexpensive, and easy-to-implement solution to these problems using an application of mixup between data and noise.

- We extend regularization penalties to larger-scale cases and propose a novel monotonicity regularization that adds more capability to the convolutional classifiers including the detection of anomalous data and the computation of visual explanation maps.

- Finally, to the best of our knowledge, this is the first work that proposes using monotonicity for controllable disentangled representation learning. Interestingly, our results showed that monotonic factorized VAEs generate data with smooth relationships between outputs and latent variables. For example, sequences of latent variables result in reconstructions with color sequences that follow the order of the HUE cycle. This property is not explicitly enforced during training and results from monotonicity penalties.

We hope you find our responses satisfactory. If you still have questions, we hope to discuss them further with you to improve our paper.

---

### Decision · Program_Chairs · 2022-01-20

**Decision:**

Reject

**Comment:**

This paper proposes a new approach to enforce monotonicity in the context of risk minimization, or to promote it as an inductive bias.  This improves upon existing point-wise gradient based methods by expanding the region where monotonicity is enforced.  Group monotonicity is found valuable as a regularization for convolutional models, and multiple applications were shown where the approach appears effective.

The paper is well written, and received detailed discussion. Despite the rebuttal, some major concerns remain, such as drop in accuracy, and empirical estimate of the probability that Definition 1 would not hold over the distribution in question.  Overall, revisions are needed to make the paper publishable.